# Development of a novel non-invasive biomarker panel for hepatic fibrosis in MASLD

Lars Verschuren [1,13] ✉, Anne Linde Mak [2,13], Arianne van Koppen [1], Serdar Özsezen[1], Sonia Difrancesco [1], Martien P. M. Caspers [1], Jessica Snabel[1], David van der Meer [3], Anne-Marieke van Dijk [2], Elias Badal Rashu [4], Puria Nabilou [4], Mikkel Parsberg Werge[4], Koen van Son[2], Robert Kleemann[1], Amanda J. Kiliaan [5], Eric J. Hazebroek[6], André Boonstra [7], Willem P. Brouwer[7], Michail Doukas [8], Saurabh Gupta[9], Cornelis Kluft[10], Max Nieuwdorp [2], Joanne Verheij[11], Lise Lotte Gluud [4], Adriaan G. Holleboom[2,14], Maarten E. Tushuizen[12,14] & Roeland Hanemaaijer[1,14]

Accurate non-invasive biomarkers to diagnose metabolic dysfunction-associated steatotic liver disease (MASLD)-related fibrosis are urgently needed. This study applies a translational approach to develop a blood-based biomarker panel for fibrosis detection in MASLD. A molecular gene expression signature identified from a diet-induced MASLD mouse model (LDLr−/−.Leiden) is translated into human blood-based biomarkers based on liver biopsy transcriptomic profiles and protein levels in MASLD patient serum samples. The resulting biomarker panel consists of IGFBP7, SSc5D and Sema4D. LightGBM modeling using this panel demonstrates high accuracy in predicting MASLD fibrosis stage (F0/F1: AUC = 0.82; F2: AUC = 0.89; F3/F4: AUC = 0.87), which is replicated in an independent validation cohort. The overall accuracy of the model outperforms predictions by the existing markers Fib-4, APRI and FibroScan. In conclusion, here we show a disease mechanism-related blood-based biomarker panel with three biomarkers which is able to identify MASLD patients with mild or advanced hepatic fibrosis with high accuracy.

Metabolic dysfunction-associated steatotic liver disease (MASLD) comprises a spectrum of diseases, ranging from metabolic dysfunction-associated isolated steatosis, in which the predominant histological characteristic is lipid accumulation in hepatocytes, to metabolic dysfunction-associated steatohepatitis (MASH), with additional hepatic injury leading to fibrosis, and may ultimately culminate in MASH-related cirrhosis and hepatocellular carcinoma (HCC) in some patients[1,2]. Obesity and insulin resistance are strongly associated with MASLD, both via increased delivery of free fatty acids to the liver and through increases of hepatic lipogenesis associated with hyperglycemia and hyperinsulinemia[3]. With the recent swift rise in

prevalence of MASLD to around 30% worldwide, an urgent need has arisen to identify those patients who develop severe consequences from this cardiometabolic liver disease[1]. Research in large cohorts has shown that hepatic fibrosis is the major determinant of overall and liver-related mortality in patients with MASLD[3–5]. At the molecular level, the development of hepatic fibrosis is a dynamic process. In the healthy liver and even at early stages of MASLD, the processes of damage and repair are in balance[6,7]. A current hypothesis is that this balance can become disturbed by repetitive hepatic injury as a result of an overload of intrahepatic fat in MASLD/MASH[7]. Injured hepatocytes trigger regenerative responses, recruiting immune cells,

myofibroblasts, and hepatic progenitor cells. When the hepatic injury is chronic and repetitive, or when the repair mechanisms become maladaptive and dysregulated, these regenerative responses derail and cause continuous deposition of collagen by myofibroblasts and a loss of pro-resolution macrophages that can remodel extracellular matrix[6]. This results in a net effect of an increase in extracellular matrix production in the liver: scarring, with hepatic fibrosis.

The current gold standard to detect and stage hepatic fibrosis is a liver biopsy, an invasive procedure that carries risks such as post-procedural bleeding and sampling error, and thus is far from ideal to screen for hepatic fibrosis in the large population of patients with MASLD. The search for non-invasive testing methods to diagnose MASLD fibrosis is currently a trending topic in research[8–10]. Various non-invasive tests have been developed to identify patients with advanced liver fibrosis. Where analysis of single liver enzymes as aspartate aminotransferase (AST) and alanine aminotransferase (ALT) fail to predict the presence of fibrosis in MASLD, extensive multi-omics studies have shown that combinations of multiple biomarkers are useful for detection of different stages of MASLD[11–13], but large number of omics biomarkers are difficult to implement in clinical practice. The most clinically used blood-based biomarker panels for MASLD fibrosis are the Fibrosis-4 (Fib-4) test, the enhanced liver fibrosis test (ELF) and the APRI (AST-to-platelet ratio index). Although these have been implemented in clinical care paths for patients with MASLD in several countries[14], they still have major drawbacks. The ELF test, which is designed on coincidental expression of proteins in the disease state[15], has limitations in its predictive value and diagnostic accuracy, e.g. in populations with low disease prevalence[16–18]. The Fib-4 and APRI test do not include direct fibrosis markers and are a calculated score based on AST and platelet count (APRI) and additionally age and ALT (Fib-4)[19,20]. Fib-4 has a high risk of overdiagnosis and false negatives[21] and limited performance to identify advanced fibrosis among subjects with MASLD[22], especially in the absence of diabetes[23]. In addition, the APRI score shows conflicting results for advanced MASH fibrosis[20,24]. The most clinically used imaging modality for advanced MASH fibrosis is transient elastography with FibroScan®, which has demonstrated an AUC of 0.83 in the LITMUS metacohort (available in $n = 632$)[24] but could potentially over-diagnosis patients as having a higher stage of fibrosis[25]. Biomarkers that are mechanistically linked to the fundamental process of MASLD-related fibrosis, i.e. the overexpression and deposition of new matrix proteins, may allow a more accurate identification and prediction of the fibrosis stages in patients[26].

In previous studies we described a time-dependent sequence of key molecular processes during the development of MASH and fibrosis in a translational murine model, which enabled us to identify an early molecular gene expression based of fibrogenesis[27,28]. In the current study we used a targeted approach aimed at establishing a panel of blood-based biomarkers that is mechanistically linked to the fibrogenic process in patients with MASLD and can be used to non-invasively and accurately diagnose the fibrosis stage in MASLD patients. To this end, this work is based on a tracer study in high-fat diet (HFD)-fed LDLr−/−.Leiden mice to identify candidate biomarkers with a mechanistic link to the formation of new collagen. We then translated these biomarkers to their human counterparts using different cohorts of patients with histologically confirmed MASLD. This resulted in a mathematical model using three blood-based biomarkers that accurately predicts the fibrosis stage of MASLD patients.

## Results

### Identification of candidate biomarkers of fibrogenesis in mice
Ldlr-/-.Leiden mice fed a high-fat diet (HFD) for 24 weeks displayed pronounced obesity in comparison to the age-matched control mice that were fed a low-fat chow diet (Fig. 1a). Moreover, plasma levels of ALT (Fig. 1b) and AST (Fig. 1c) as well as the liver weight were increased in HFD-fed mice in comparison with the chow-fed mice at sacrifice.

Biochemical analysis of hepatic lipids revealed a significant increase in cholesterol esters (Fig. 1d), free cholesterol (Fig. 1e) and triglycerides (Fig. 1f) in HFD-fed animals. Histological examination of hepatic steatosis (Fig. 2a), inflammation (Fig. 2b), and perisinusoidal fibrosis (Fig. 2c) showed development of hepatic inflammation and fibrosis in the HFD-fed mice.

To examine the genes and pathways altered in response to HFD-induced MASH and fibrosis, we conducted global gene expression analysis. Our findings demonstrate significant alterations in gene expression between the livers of HFD-fed versus chow-fed mice. Specifically, we identified 2696 genes that exhibited statistically significant differences (FDR < 0.001). Out of these genes, 1048 were found to be downregulated, while 1648 were upregulated following 24 weeks of HFD treatment. Subsequent enrichment analysis of these differentially expressed genes (DEGs) revealed an overrepresentation of pathways associated with lipid metabolism, inflammation, and fibrosis (Fig. 3a). Notably, the Hepatic Fibrosis Signaling Pathway stood out as one of the most significantly affected pathways ($p$-value < 0.0001; Fig. 3c). Next, we evaluated the impact of 24 weeks of HFD feeding on liver fibrosis at the protein level by analyzing protein turnover rates using deuterated water labeling. Our dynamic proteomics analysis was focused on the production of fibrillar collagens, viz. Collagen1α1 and Collagen1α2 from the guanidine-soluble and guanidine-insoluble fractions, and Collagen3α1 from the guanidine-insoluble fraction. HFD feeding resulted in significantly more $D_2O$-labeled hepatic fibrillar collagens as compared to chow-fed animals, (Fig. 3b) indicating enhanced fibrogenesis in mice fed HFD for 24 weeks.

Correlation analysis revealed that 1112 DEGs correlated significantly with the fractional synthesis rate of at least one of the selected fibrillar collagens (Supplementary Data 1). We prioritized genes based on their minimum absolute correlation coefficient and selected 645 DEGs with a correlation coefficient greater than 0.9 with the fractional synthesis rate of at least two analyzed collagens. Thus, these genes strongly correlate to the synthesis of at least two collagens and could serve a candidate set of biomarkers for fibrosis deposition.

### Verification of candidate markers on gene level in human liver biopsies of translation cohort
To evaluate which of these genes were also regulated in liver tissue of patients with MASLD, we collected formalin-fixed paraffin-embedded liver biopsy material from patients with MASLD as a translation cohort. For this cohort, sixty-seven archived liver biopsy samples were selected for gene expression analysis. The average age of the patients was 49 years, 62% of patients was male, the average BMI was 29.9 kg/m² and 32% had type 2 diabetes mellitus. Liver biopsies showed various stages of fibrosis (13%-F0, 25%-F1; 37%-F2; 15%-F3; 9%-F4), see Supplementary Data 2.

RNA sequencing analysis on these liver biopsies showed relatively low DEGs in patients with MASLD fibrosis stage F1 and F2 (629 and 1049 DEGs, $p < 0.01$) as compared to biopsies classified as fibrosis stage F0. In biopsies classified as fibrosis stage F3 and stage F4 the number of DEGs as compared to stage F0 was larger (5016 and 6299, respectively) (Fig. 4a). Gene set enrichment analysis indicated regulation of several pathways related to MASLD and hepatic fibrosis, including lipid metabolism, inflammation, and hepatic fibrosis signaling. The biopsies of patients with stage F3 or F4 fibrosis, as compared to biopsies with fibrosis stage F0, showed high similarity in differentially expressed pathways (Fig. 4b), with one of the most significant pathways being Hepatic Fibrosis Signaling (Fig. 4c) indicating the majority of genes are upregulated in patients with F4 stage fibrosis as compared to F0 stage. In order to identify the most promising candidate biomarkers, a comparative gene expression analysis was performed for the 645 mouse signature genes that strongly correlate to the synthesis of collagens with significant genes in human patients (DEGs in fibrosis stage F3 or F4 vs. no fibrosis). This analysis revealed

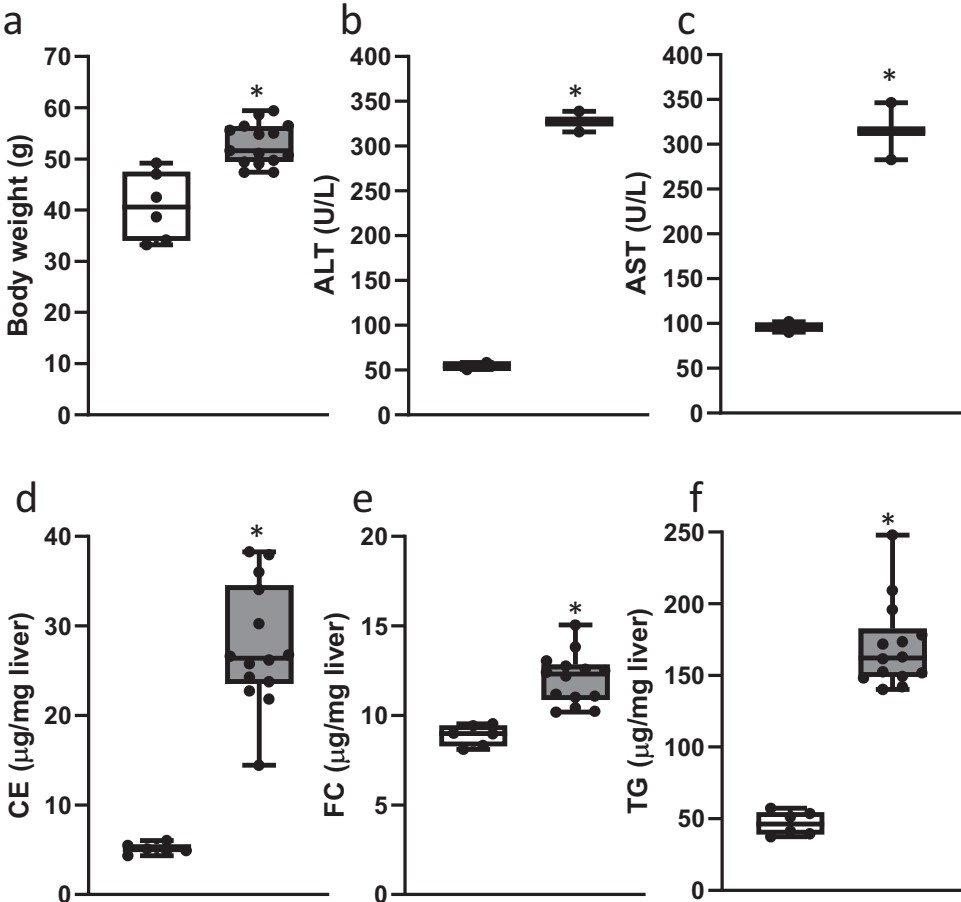

**Fig. 1 | HFD feeding in LDLr−/−.Leiden mice results in obesogenic phenotype.**
**a** Body weight in chow-fed (white bars; $n = 6$) and HFD-fed (gray bars; $n = 15$) LDLr −/−.Leiden mice after 24 weeks ($P = 0.025$); **b** ALT levels (pooled plasma of 3 mice; $n = 2$ for chow; $n = 2$ for HFD; $P = 0.002$); **c** AST levels (pooled plasma of 3 mice; $n = 2$ for chow; $n = 2$ for HFD; $P = 0.02$); **d** Hepatic cholesterol ester concentration ($n = 6$ for chow; $n = 14$ for HFD; $P < 0.0001$); **e** Hepatic free cholesterol concentration ($n = 6$ for chow; $n = 14$ for HFD; $P < 0.0001$); **f** Hepatic triglyceride concentration ($n = 6$ for chow; $n = 14$ for HFD; $P < 0.0001$). Values are shown in box and whisker plots, data are median (horizontal line), interquartile range (boxes), and min. to max. (error bars); Data consist of biological replicates where HFD-fed mice are compared to control chow-fed mice. A two-sided Student's $t$-test was used to test the statistical significance; *$p < 0.01$.

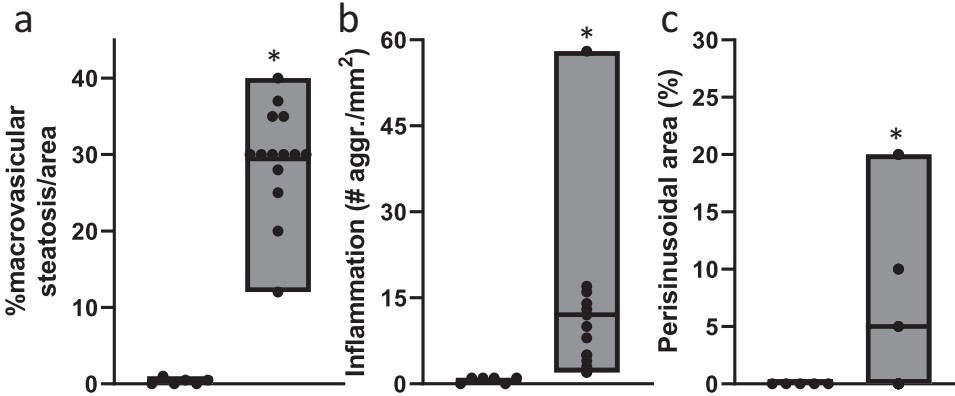

**Fig. 2 | HFD feeding induces liver pathology associated to MASH.**
**a** Macrovesicular steatosis in chow-fed (white bars; $n = 6$) and HFD-fed (gray bars; $n = 14$) LDLr−/−.Leiden mice after 24 weeks ($P < 0.0001$); **b** Number of inflammatory aggregates ($n = 6$ for chow; $n = 14$ for HFD; $P = 0.01$); **c** Perisinusoidal fibrosis area ($n = 5$ for chow; $n = 14$ for HFD; $P = 0.02$). Values are shown as floating bars (min. to max.), line indicates mean. Data consist of biological replicates where HFD-fed mice are compared to control chow-fed mice. A two-sided Student's $t$-test with Welch's correction was used to test the statistical significance; *$p < 0.05$.

that among 284 DEGs in F3 classified biopsies, 86% ($n = 244$) showed regulation in the same direction as the mouse DEGs (Fig. 4d), and among the 352 DEGs in F4 classified biopsies, 91% ($n = 321$) showed regulation in the same direction as the mouse DEGs (Fig. 4e). Based on these data, a subset of 238 genes, showed a remarkable overlap across both fibrosis stage F3 and fibrosis stage F4. This overlap in gene regulation was found to be in correspondence with the regulatory trends observed in the signature genes of the mouse experimental dataset.

Next, we investigated whether the identified 238 fibrosis biomarker candidate were translated into a circulating protein. To define

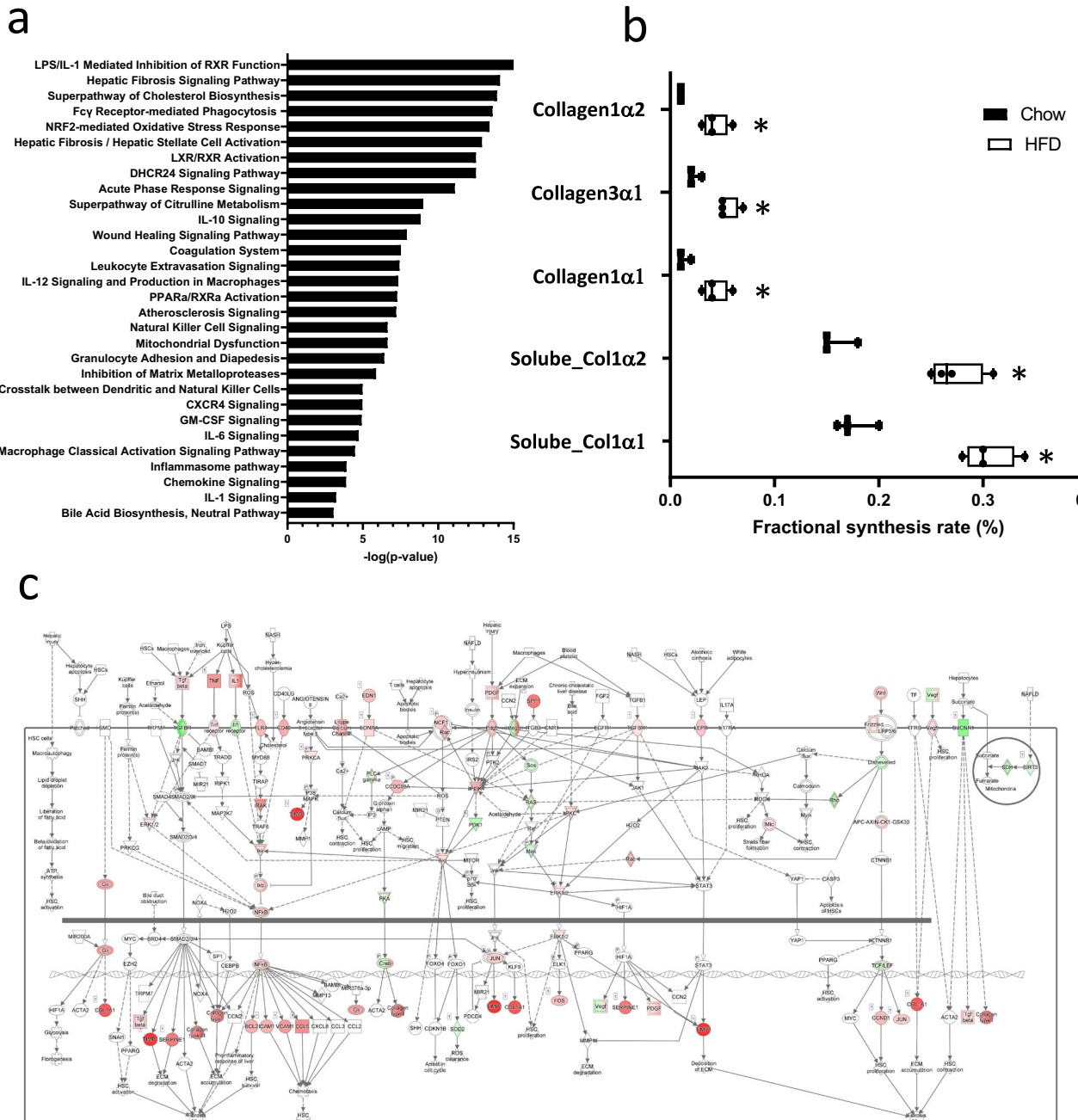

**Fig. 3 | Molecular effects in liver tissue. a** Top selected pathways affected in HFD-fed mice as compared to chow-fed mice. **b** Box and Whisker plot showing fractional synthesis rate of collagens in liver of chow- or HFD-fed mice (collagen1α2, $P = 0.007$; collagen3α1, $P = 0.004$; collagen1α1, $P = 0.014$; soluble collagen1α2, $P = 0.001$; soluble collagen1α2, $P = 0.001$) after 24 weeks of treatment ($n = 3$ chow and $n = 4$ HFD). Data are median (vertical line), interquartile range (boxes) and 10–90% percentile (error bars). A two-sided Student's t-test was used to test the statistical significance; *$p < 0.01$. **c** Visualization of significant genes up- (red) and down (green) regulated in the Hepatic Fibrosis Pathway in HFD-fed mice as compared to chow-fed control mice.

such proteins, we used information from the Cortellis Drug Discovery Intelligence database (Clarivate, https://www.cortellis.com/), a human biomarker atlas, and conduct a comprehensive literature search. Consequently, this methodology yielded a curated set of 21 candidate blood-based biomarkers to be futher validated on protein level in serum samples of the translation cohort.

### Verification of candidate markers on protein level in human serum samples

Twenty-one biomarker assays were tested, optimized and validated, as detailed in the methods section, to assess their sensitivity and performance for the analysis of serum samples. Out of the 21 assays, 10

assays did not meet the criteria, resulting in a selection of 11 biomarker assays that were used to measure serum samples of the translation cohort. The obtained data demonstrated that the candidate biomarkers can be reliably detected, and further reveal an elevated concentration of certain biomarkers in samples exhibiting an increased fibrosis score (Supplementary Fig. 2).

### Serum biomarker verification in an independent testing cohort

The next step involved the assessment of serum protein levels for the potential biomarkers in an independent testing cohort, which consisted of a total of 128 patients with histologically confirmed MASLD. The average age was 49 years, 64% of patients were male, the average

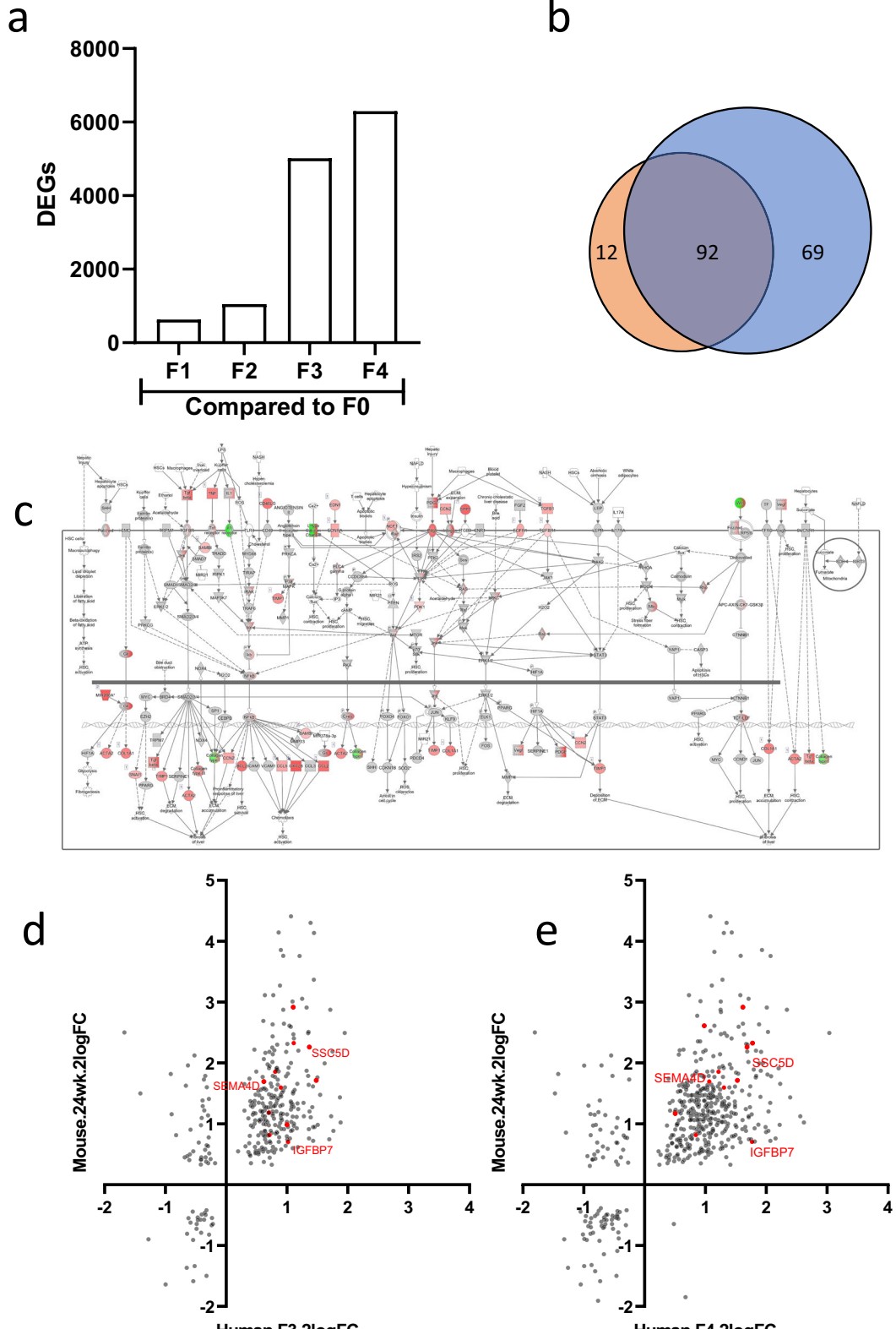

**Fig. 4 | Translational gene markers in human biopsies. a** Quantification of differentially expressed liver genes in MASLD patients across various fibrosis stages ($P < 0.01$). **b** Comparative analysis of pathway overlaps between stages F3 (orange) and F4 (blue) against stage F0. **c** Visualization of the Hepatic Fibrosis Signaling Pathway using DEGs in F4 patients as compared to F0 ($P < 0.01$). **d, e** Dual-axis log2-FoldChange plots showing signature genes that exhibit differential expression in both mouse models (24 weeks, HFD vs. chow) and human MASLD patients (stages F3 and F4 vs. F0). The 11 selected biomarkers are highlighted as red-colored genes with the selection of 3 biomarkers specifically named.

**Table 1 | Clinical characteristics of the testing cohort**

| Demographics | N = 128 |
|---|---|
| Age [years], mean (SD) | 49 (13) |
| Male, n (%) | 72 (56%) |
| Clinical characteristics | |
| BMI, mean (SD) | 34.2 (6.0) |
| T2D diagnosis, n (%) | 52 (40.6%) |
| Fasting glucose [mmol/L], mean (SD) | 6.2 (1.2) |
| HbA1c [mmol/mol], mean (SD) | 41.7 (9.5) |
| AST, mean (SD) | 61.0 (53.0) |
| ALT, mean (SD) | 75.6 (61.0) |
| GGT, mean (SD) | 120.0 (203.2) |
| Thrombocytes, mean (SD) | 224.0 (88.3) |
| Liver histology scoring | |
| Steatosis grade, n (%) | |
| 0 | 19 (14.8%) |
| 1 | 46 (35.9%) |
| 2 | 35 (27.3%) |
| 3 | 28 (21.9%) |
| Lobular inflammation grade, n (%) | |
| 0 | 10 (7.9%) |
| 1 | 104 (81.9%) |
| 2 | 13 (10.2%) |
| Ballooning, n (%) | |
| 0 | 55 (43.0%) |
| 1 | 51 (39.8%) |
| 2 | 22 (17.2%) |
| NAS score, n (%) | |
| 0 | 1 (0.8%) |
| 1 | 19 (15.0%) |
| 2 | 23 (18.1%) |
| 3 | 23 (18.1%) |
| 4 | 29 (22.8%) |
| 5 | 20 (15.7%) |
| 6 | 10 (7.9%) |
| 7 | 2 (1.6%) |
| F-score, n (%) | |
| 0 | 7 (5.5%) |
| 1 | 31 (24.2%) |
| 2 | 38 (29.7%) |
| 3 | 27 (21.1%) |
| 4 | 25 (19.5%) |

BMI was 34.2 kg/m$^2$ and 40% had type 2 diabetes mellitus. Liver histology scores and clinical characteristics are given in Table 1, including the average serum concentrations of the biomarkers in this cohort in Table 2.

In order to test the capability of our candidate biomarkers in detecting hepatic fibrosis, a machine-learning model was constructed using Light Gradient Boosting Machine (LGBM). This model was first used for feature selection of biomarkers with the highest contributory effect on the prediction (Fig. 5). The results indicated that the three best-performing features were Insulin-like growth factor-binding protein 7 (IGFBP7; Fig. 6a), Scavenger Receptor Cysteine Rich Family Member With 5 Domains (SSc5D; Fig. 6b) and Semaphorin 4D (Sema4D; Fig. 6c). Next, the model was trained to classify patients into three primary fibrosis categories: no fibrosis (F0 and F1; n = 38), mild fibrosis (F2; n = 38), and advanced fibrosis (F3 and F4; n = 52). First,

samples were divided into a balanced training set (n = 30 no fibrosis; n = 30 mild fibrosis; n = 30 advanced fibrosis) and a holdout test set (n = 8 no fibrosis; n = 8 mild fibrosis; n = 22 advanced fibrosis). A ten-fold cross-validation was applied to evaluate the overall prediction accuracy of the model. The training model had an overall AUC of 0.87 ± 0.083 based on all subgroups and its overall accuracy was 0.744 ± 0.071. After selecting the top 3 performing components for our non-invasive blood biomarker panel, we tested the model by defining its accuracy in distinguishing the various fibrosis stages (no fibrosis, mild fibrosis and advanced fibrosis). The model achieved an area under the receiver operating curve (AUROC) of 0.82 for fibrosis stage F0/F1, a robust AUROC of 0.89 for fibrosis stage F2, and a strong AUROC of 0.87 for distinguishing fibrosis stages F3 and F4 (Fig. 6d). The model's effectiveness in discriminating fibrosis categories was further evaluated by computing sensitivity and specificity metrics, both derived from the associated confusion matrix (Fig. 6e, f). In addition to the top-performing biomarkers, clinical variables such as type 2 diabetes, sex, and BMI were incorporated in the model to assess whether these covariates contribute to the model's performance. Our data showed that none of these clinical variables resulted in a significantly improved prediction of fibrosis in the testing cohort (Supplementary Fig. 3).

### Serum biomarker validation in a second independent cohort

The final step was to perform an external validation of the LGBM model, hereafter referred to as the TLM3 model (TNO LGBM MASLD model, based on the three biomarkers). For this, we analyzed the biomarkers in serum of patients with histologically confirmed MASLD (n = 156). The average age was 55.8 years, 53.8% of patients was male, the average BMI was 34.75 kg/m$^2$ and 49.3% had type 2 diabetes mellitus. Liver histology scores and clinical characteristics are given in Table 3. The average serum concentrations of the biomarkers in this cohort are shown in Fig. 7a–c. To illustrate the dynamics of the biomarkers over the patient subgroups in the independent validation cohort, we also included data from the testing cohort. The confusion matrix (Fig. 7d) and the AUROC curve (Fig. 7e) show the model performance in discriminating fibrosis subgroups (F0/F1 vs. F2 vs. F3/F4) in the validation cohort. These data confirm the predictive accuracy of the model showing an AUC of 0.84 for both the F0/F1 and F3/F4 subgroups. The prediction of the F2 subgroup was more modest in this independent validation cohort from Denmark as compared to the Dutch testing cohort. We subsequently evaluated the performance of our TLM3 model against three known NITs: Fib-4, APRI, and FibroScan in the independent validation cohort. The overall accuracy of the TLM3 model outperformed Fib-4, APRI, and FibroScan (Fig. 7f). While the TLM3 model showed consistent robustness across various metrics, it was particularly noteworthy in its predictive capabilities for both early (F0/F1) and advanced (F3/F4) fibrosis stages (Supplementary Fig. 4). To perform a more comprehensive analysis, a comparison between Fib-4 and TLM3 was made across the combined patient cohorts, which included the training set, the first validation set, and the second validation set (Supplementary Table 2). The analysis shows that TLM3 consistently outperformed Fib-4 across all performance metrics for liver fibrosis stages F2 and F3/F4. For the early stages of fibrosis (F0/F1), the specificity and sensitivity of the two biomarkers were nearly identical (0.70 vs. 0.69 and 0.83 vs. 0.82, respectively). TLM3 significantly exceeds Fib-4 prediction in terms of precision. In all, this comprehensive evaluation demonstrates enhanced diagnostic efficacy of our TLM3 model over Fib-4, APRI, and FibroScan.

## Discussion

In this study, we present the development and validation of a reliable blood-based biomarker panel that accurately predicts the fibrosis categories F0-F1, F2, and F3–F4 in patients with MAFLD using only three biomarkers that are molecularly linked to the active collagen

**Table 2 | Average biomarker concentrations of the patient cohort**

|  | F0–F1 |  | F2 |  | Pval | F3–F4 |  | Pval |
|---|---|---|---|---|---|---|---|---|
|  | Average | stdev | Average | stdev | vs F0–F1 | Average | stdev | vs F0–F1 |
| IGFBP7 (ng/ml) | 183.5 | 33.3 | 210.1 | 49.4 | 0.008 | 363.8 | 198.5 | 0.000 |
| SSC5D (ng/ml) | 1.9 | 1.5 | 5.6 | 2.9 | 0.000 | 13.0 | 11.1 | 0.000 |
| SEMA4D (µg/ml) | 0.8 | 0.3 | 1.1 | 0.2 | 0.000 | 1.3 | 0.7 | 0.000 |
| TNC (ng/ml) | 11.1 | 4.0 | 8.3 | 2.5 | 0.001 | 12.7 | 7.7 | 0.239 |
| PLAU (pg/ml) | 895.1 | 673.3 | 988.0 | 416.7 | 0.476 | 1346.9 | 774.7 | 0.005 |
| CXCL10 (pg/ml) | 166.1 | 76.2 | 174.5 | 64.5 | 0.612 | 208.9 | 92.3 | 0.024 |
| THBS1 (µg/ml) | 23.7 | 17.2 | 27.0 | 20.6 | 0.447 | 35.8 | 22.5 | 0.007 |
| PAM (ng/ml) | 91.1 | 36.5 | 83.4 | 30.7 | 0.323 | 85.4 | 43.6 | 0.517 |
| VCAN (ng/ml) | 103.1 | 18.5 | 103.8 | 31.7 | 0.904 | 98.8 | 44.9 | 0.579 |
| ADAMTS2 (ng/ml) | 8.7 | 6.8 | 16.4 | 8.9 | 0.000 | 24.1 | 14.7 | 0.000 |
| FBN1 (ng/ml) | 19.9 | 14.9 | 27.5 | 12.8 | 0.020 | 41.0 | 21.4 | 0.000 |

Values are expressed as concentration in g/ml. A two-sided Student's *t*-test was used to test the statistical significance.

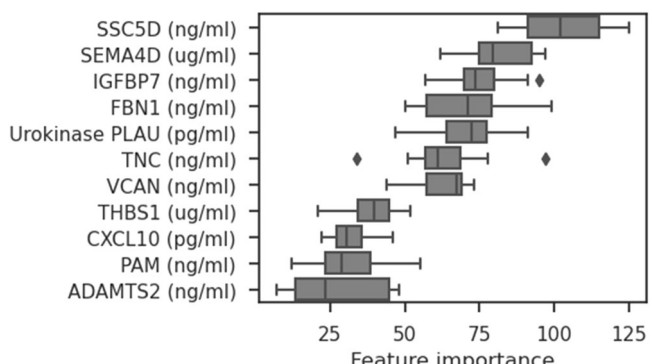

**Fig. 5 | Biomarker selection.** Box and Whisker plot showing feature importance list of the machine learning model, showing the contribution of each of the 11 serum biomarkers to the classification into fibrosis groups in 128 patients of the testing cohort. Data are median (vertical line), interquartile range (boxes) and 10–90% percentile (error bars). The diamonds indicate outliers.

turnover process in the liver. This biomarker panel allows to non-invasively identify patients who would benefit most from specialized clinical care. To achieve this, we adopted a mechanism-based translational approach to identify candidate biomarkers that are mechanistically linked to the fibrogenic process in a translational mouse model[28]. We determined human relevance by comparison analysis of hepatic transcription levels, ensuring that relevant fibrosis pathways in MASLD were abundant in both the mouse model and the human translation cohort. Serum samples from the human translation cohort were subsequently used for the identification of blood-based biomarkers on the protein level. In an independent testing cohort, a model employing only three blood-based biomarkers showed a high degree of accuracy and allowed identification of clinically relevant fibrosis categories in MASLD. Validation of the LGBM model, hereafter referred to as TLM3, was performed in samples from MASLD patients in an independent validation cohort from Denmark. We successfully replicated the high-performance predictions in the testing cohort and showed that the overall accuracy of our model outperformed Fib-4, APRI and FibroScan predictions.

The starting point was the LDLr−/−.Leiden mouse model which closely mimics human pathophysiology and responds effectively to exercise, nutritional, and pharmaceutical treatments[27–29]. The translational value of the LDLr−/−.Leiden mouse model in relation to human biopsies was established through a direct comparison with the C57BL/6 mouse model, fed a Western-type diet, as illustrated in Supplementary Fig. 5. This comparison revealed that the LDLr−/−.Leiden mouse

model shares a much higher overlap with human genes than the C57BL/6 mouse model. This observation aligned with the previous data[27–29] that influenced our decision to choose the HFD-fed LDLr −/−.Leiden mouse model for further analysis. Dynamic proteomics analysis in mouse livers, using $D_2O$-tracers, enabled the identification of newly synthesized extracellular matrix proteins, including collagens. Correlation analysis of the proteomics data with hepatic transcription levels identified a gene-based fibrosis signature which is closely associated with new collagen formation. Correlation analyses that focus on the static mRNA levels of Col1a1 are frequently used for identifying biomarkers. In our study, however, we identified biomarkers by correlating them with dynamic proteomics. This approach resulted in novel biomarkers that would not have been identified by direct correlation with Col1a1 mRNA expression, either in mouse or in human studies. Specifically, this approach led to the selection of 6 out of 11 biomarkers that would not have been selected based correlation with mouse Col1a1 expression, and none of the biomarkers would have been identified using correlation with human Col1a1 mRNA expression (see Supplementary Table 1).

A rigorous approach was taken to translate murine candidate biomarkers to humans, starting by translating the mouse hepatic transcriptome responses to hepatic transcription levels in a cohort of patients with MASLD across fibrosis stages (translation cohort). We focused on candidate genes showing consistent differential expression between no fibrosis and advanced fibrosis in both mice and humans. Following this, genes encoding serum proteins were selected, and a pragmatic assortment of serum proteins amenable to detection through enzyme-linked immunosorbent assay (ELISA) assays yielded a set of 11 promising candidate biomarkers. Machine Learning has been shown to be an eminently suitable tool to translate the protein levels into predictions[30]. Here, we applied a LightGBM (LGBM) model, along with its feature selection analysis, to demonstrate the ability to predict fibrosis optimally using a panel consisting of three important biomarkers, viz. IGFBP7, SSc5D and SEMA4D.

The three biomarkers, directly correlated with the mechanism underlying dynamics of collagen deposition, each play a different role in the regulation and development of fibrosis. Insulin-like growth factor-binding protein 7 (IGFBP7) regulates insulin-like growth factors (IGFs) and modulates IGF binding to its receptors[31]. IGFBP7 has previously been identified as a potential biomarker for various cancer types, including hepatocellular carcinoma[32]. In the context of liver fibrosis in MASLD, IGFBP7 contributes to fibrogenesis by playing a role in the activation and transdifferentiation of hepatic stellate cells[33]. Hepatic IGFBP7 expression was found to be consistently increased in advanced liver fibrosis, as observed in both pediatric and adult comparative cohorts[34]. Knockdown of IGFBP7 in a mouse model of MASLD

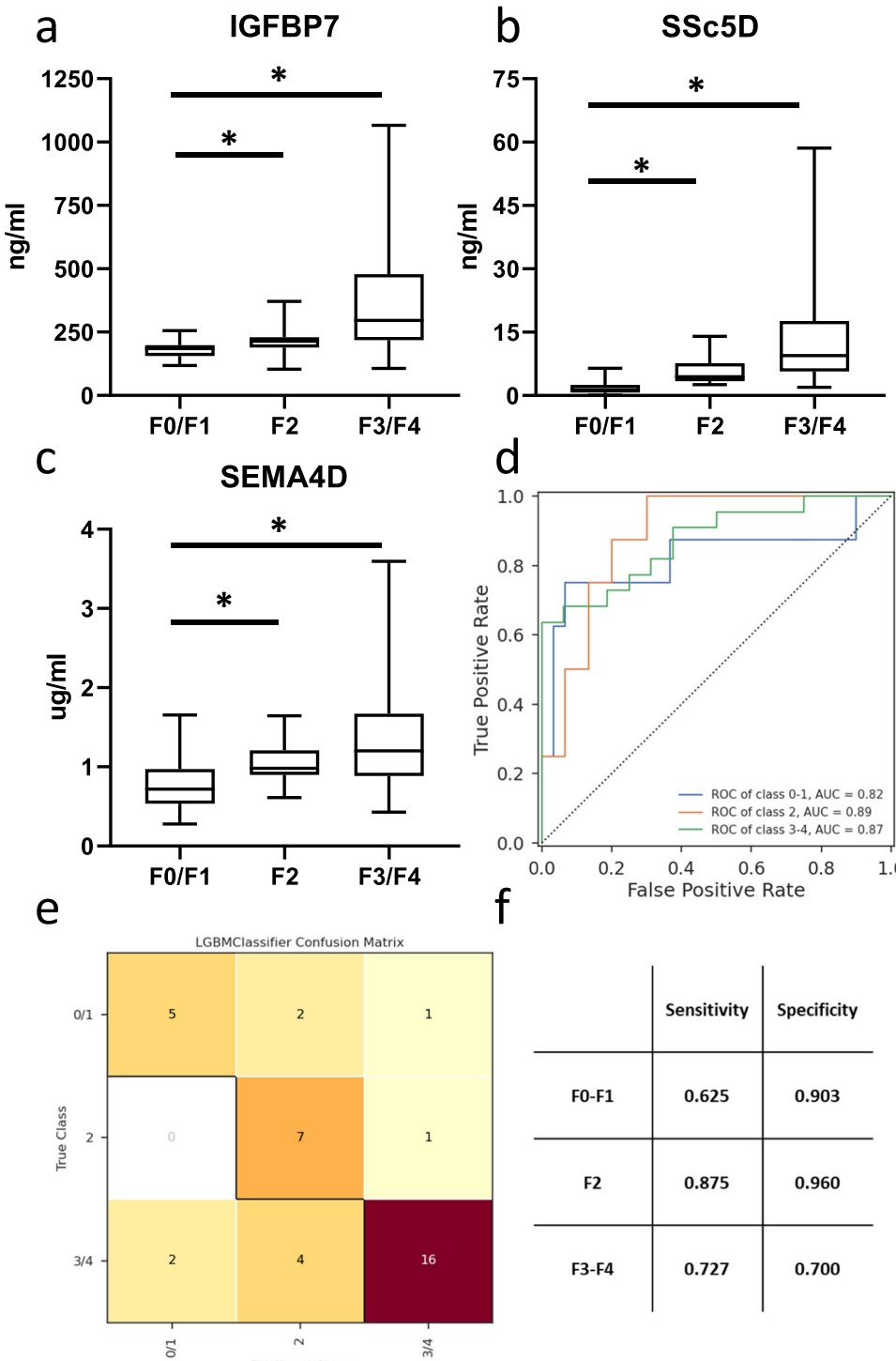

**Fig. 6 | Biomarker verification in an independent cohort.** Serum levels of **a** IGFBP7, **b** SSc5D, **c** SEMA4D in samples from 128 patients with MASLD fibrosis score (F0–F4) in testing cohort (F0/F1, $n = 38$; F2, $n = 38$; F3/F4, $n = 52$). Values from biological replicates are shown in box and whisker plots, data are median (horizontal line), interquartile range (boxes) and 10–90% percentile (error bars). A two-sided Student's t-test was used to test the statistical significance. *$p < 0.05$, vs. F0/F1 samples. **d** AUROC curve to show the predictive value of this set of biomarkers to distinguish the individual MASLD fibrosis F-scores (F0–F1 versus F2 versus F3–F4); **e** Confusion matrix of the hold-out set ($n = 8$ F0/F1; $n = 8$ F2; $n = 22$ F3/F4) of predicted and true classes; **f** Sensitivity and specificity as calculated from the LGBM classifier Confusion Matrix.

resulted in a reduction of insulin resistance and hepatic steatosis[35]. Soluble Scavenger with 5 Domains (SSc5D) is a soluble receptor that is expressed by macrophages and T cells[36]. There is a scarcity of literature on the role of SSc5D as a fibrosis biomarker. However, a recent study on SSc5D in heart failure found that the *Ssc5d* gene is also expressed by cardiac fibroblasts, and serum SSc5D concentrations were increased in patients with heart failure compared to healthy individuals[37]. Further research is required to explore its mechanism of action and broader implications in various contexts. Semaphorin-4D (Sema4D or CD100) is a transmembrane homodimeric glycoprotein member of the semaphorin family[38]. Shedding of membrane-bound Sema4D, either spontaneously or as a through proteolytic cleavage by matrix

**Table 3 | Clinical characteristics of the validation cohort**

| Demographics | n = 156 |
|---|---|
| Age [years], mean (SD) | 55 (13) |
| Male, n (%) | 84 (53.8) |
| Clinical characteristics | |
| BMI, mean (SD) | 34.75 (6.88) |
| T2D diagnosis, n (%) | 77 (49.3) |
| HbA1c [mmol/mol], mean (SD) | 46.59 (14.19) |
| AST, mean (SD) | 55.8 (38.8) |
| ALT, mean (SD) | 65.3 (63.2) |
| Thrombocytes, mean (SD) | 229.5 (73.0) |
| Liver histology scoring | |
| F-score, n (%) | |
| 0 | 38 (23%) |
| 1 | 31 (19%) |
| 2 | 30 (19%) |
| 3 | 30 (19%) |
| 4 | 33 (20%) |

metalloproteases (MMPs), releases soluble Sema4D, which has been proposed as a potential biomarker in several autoimmune or inflammatory diseases[39]. A recent prospective case-control study demonstrated significantly higher Sema4D levels in patients with MASLD compared to healthy controls, with Sema4D levels increasing with advancing fibrosis stages[40].

The combination of these three biomarkers in a panel for MASLD enabled the LGBM model to classify patients with advanced hepatic fibrosis (F ≥ 3) with high accuracy, with an AUC of 0.87. Moreover, the model could also effectively classify patients with fibrosis in an earlier stage (F2) with an AUC of 0.89. Next, the validation of the model in an independent second patient cohort (n = 156) demonstrate that the prediction of the TLM3 model could successfully be replicated for the F0/F1 and F3/F4 fibrosis stages with an AUC of 0.84. The model displayed a modest accuracy in predicting F2 fibrosis (AUC 0.62) in this cohort, which was also observed for the other NITs, e.g. Fib-4, APRI, and FibroScan. Direct comparison in the same cohort of the TLM3 model with other NITs (Fib-4, APRI and FibroScan) showed that our model showed better overall accuracy as compared to the other NITs. Future investigation will focus on combining various other NITs with our TLM3 model to further enhance the diagnostic power.

While the testing cohort was not designed for subclassification into clinical subgroups, the addition of clinical variables including type 2 diabetes, BMI, or sex did not improve the panel's performance, indicating its robustness across clinically diverse patient groups. The presence of HCC did not influence the performance of the panel either. Considering that HCC or cirrhosis is often an exclusion criterion in biomarker studies, these findings further support the biomarker panel's broad applicability in diverse patient populations, however, further validation in clinical diverse populations are needed.

While our study holds significant promise, it is important to acknowledge its limitations. This work represents the initial steps in the translational development of a mechanism-based biomarker panel, and the evaluation of its accuracy in our testing cohort should be considered exploratory. The mechanistic basis of this biomarker panel, involving the process of active extracellular matrix remodeling during fibrogenesis, could not be tested in the single-biopsy cohorts investigated in this study. Future investigations using double-biopsy prospective cohorts or intervention trials will be essential to assess the panel's performance in diagnosing ongoing fibrogenesis, which will be crucial for identifying patients who rapidly progress in fibrosis stage and will most likely benefit from pharmacological intervention. These

future steps will provide further valuable insights into the clinical utility of this biomarker panel.

In conclusion, this study describes a novel approach in successfully identifying and translating a blood-based biomarker panel for non-invasive diagnosis and staging of hepatic fibrosis in patients with MASLD. By connecting the intricate process of extracellular matrix remodeling in MASLD-related fibrosis with the development of clinical biomarkers, we have devised a promising and robust biomarker panel that directly correlates with collagen deposition. Future studies involving prospective cohorts and intervention trials will offer deeper insights into the panel's potential to detect ongoing fibrosis and guide treatment strategies in the care of MASLD patients. The potential of non-invasively assessing liver fibrosis holds significant promise for clinical application in the continuously growing population of patients with MASLD across the globe.

## Methods

### Animal studies
Twelve-week-old male LDL-receptor knockout.Leiden (LDLr−/−.Leiden) mice were obtained from the breeding facility at TNO Metabolic Health Research (Leiden, Netherlands). The mice were assigned to either a standard rodent chow diet (Sniff-R/M-V1530; Uden, Netherlands; n = 6) or a high-fat diet (HFD) (D12451; Research Diets, Inc, New Brunswick, NJ; n = 15) for a duration of 24 weeks. One group of mice (n = 15) was sacrificed before the start of the diets to define the starting condition. The mice were housed in a temperature-controlled room on a 12-h light/dark cycle in a specified pathogen-free animal facility with unlimited access to food and water. Mice were sacrificed after 24 weeks of HFD or chow feeding. One week before sacrifice, an intraperitoneal injection of deuterated water (35 mL/g body weight) was administered and 8% deuterated water was added to the drinking water to enable dynamic proteomics analysis[27]. The studies were approved by an independent Animal Welfare Body (IVD TNO; approval numbers DEC-3553) under project licenses granted by the Netherlands Central Authority for Scientific Procedures on Animals (CCD; project license numbers AVD5010020172064 and AVD5010020172931). An overview of the steps taken in the biomarker identification approach is highlighted in Supplementary Fig. 1, created with BioRender.com.

### Biochemical analysis and liver lipid analysis
Plasma ALT and AST were measured using a spectrophotometric activity assay (Reflotron-Plus, Roche). Liver lipids were extracted from liver homogenates using the Bligh and Dyer method[41] and separated by high-performance thin layer chromatography (HPTLC) on silica gel plates. Lipid spots were stained with color reagent (5 g of $MnCl_2 \cdot 4H_2O$, 32 ml of 95−97% $H_2SO_4$ added to 960 ml of $CH_3OH:H_2O$ 1:1 v/v) and triglycerides, cholesteryl esters and free cholesterol were quantified using TINA version 2.09 software (Raytest, Straubenhardt, Germany).

### Histological assessment of MASH in LDLr−/−.Leiden
Histological changes were assessed in haematoxylin and eosin-stained liver sections (3 µm thick). Steatosis was determined at a ×40 magnification by analyzing the percentage of the total liver slice area affected. Hepatic inflammation was analyzed by counting the number of inflammatory foci per field at a ×100 magnification (view size 3.1 mm²) in five different fields per specimen, and was expressed as the average number of foci per field. Hepatocyte ballooning is sporadically observed in this model and was not quantitatively scored. Hepatic collagen content was stained histochemically using Picro-Sirius Red staining (Chroma, WALDECK-Gmbh, Münster, Germany).

### Dynamic targeted protein analysis
A dynamic proteomics platform[42] was used to quantify the fractional synthesis rates of a specific set of extracellular matrix proteins (ECM) using stable isotope labeling and a liquid chromatography-mass

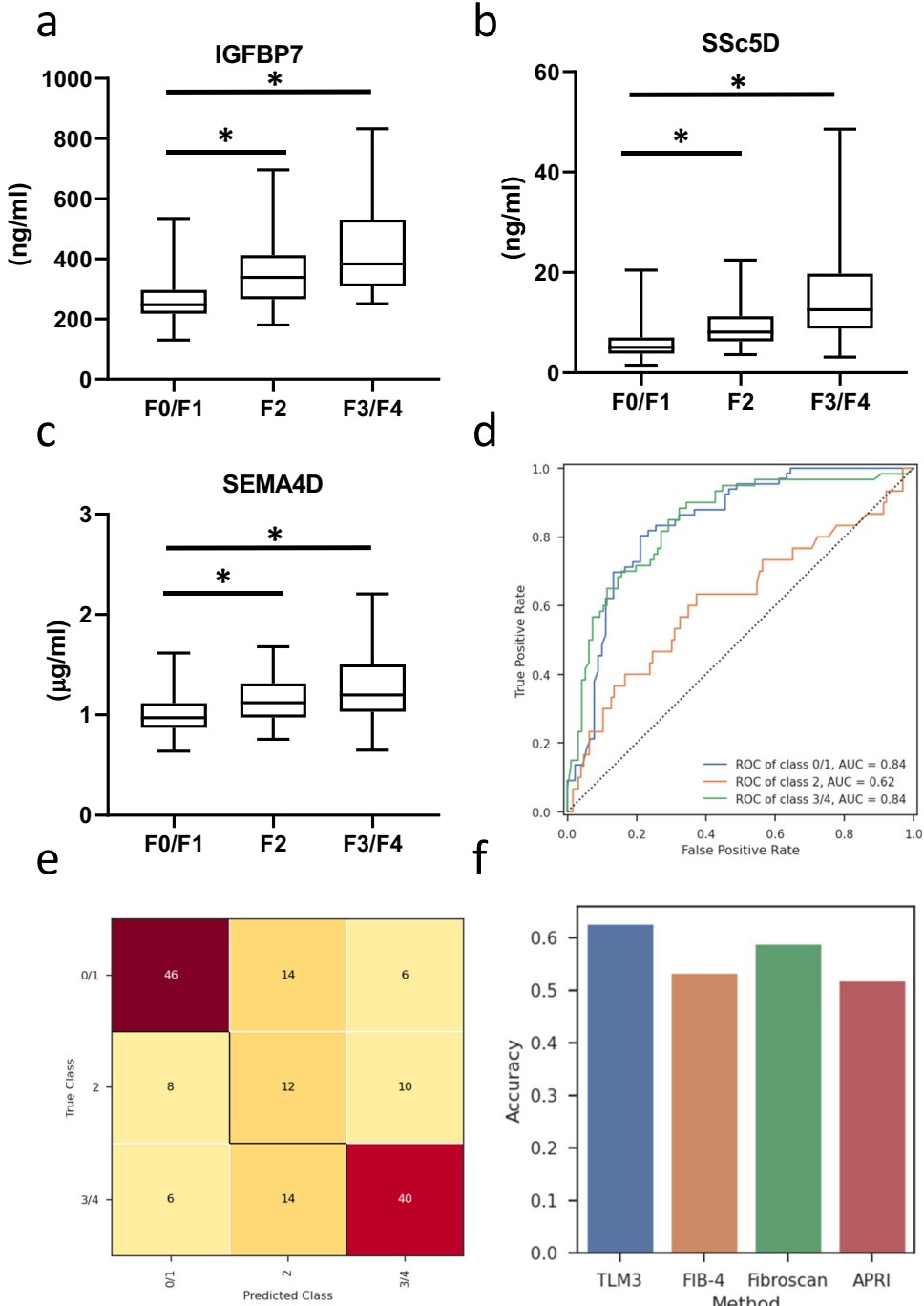

**Fig. 7 | Biomarker validation in a second independent cohort.** Serum levels of **a** IGFBP7, **b** SSc5D, **c** SEMA4D as measured in samples from 156 patients from the independent validation cohort from Denmark (F0/F1, *n* = 66; F2, *n* = 30; F3/F4, *n* = 60). Values are shown in box and whisker plots, data are median (horizontal line), interquartile range (boxes) and 10–90% percentile (error bars). A two-sided Student's *t*-test was used to test the statistical significance. *\*p* < 0.05, vs. F0/F1 samples. **d** AUROC curve to show the predictive value of this set of biomarkers to distinguish the individual MASLD fibrosis F-scores (F0/F1 versus F2 versus F3/F4). **e** Confusion matrix of the hold-out set of predicted and true classes. **f** Overview of model performances (accuracy) of different NITs in predicting fibrosis.

spectrometry (LC-MS)-based mass isotopomer analysis. Briefly, after receiving deuterated water for 7 days, mice were sacrificed and cellular, guanidine-soluble ECM proteins and residual insoluble ECM proteins was extracted and fractionated from liver tissue. The follow-up analysis was focused on collagen1α1 and collagen1α2 in the guanidine soluble fraction and collagen1α1, collagen1α2 and collagen3α1 in the insoluble fraction. The protein fractional synthesis rates (the fraction of each newly synthesized protein during the 7-day labeling period) were calculated using mass isotopomer analysis as previously described[43].

## Selection of translation cohort

To translate the murine findings to humans, 74 patients with MASLD were selected of whom stored liver biopsy material was available at the pathology department of the Erasmus Medical Center and Utrecht Medical Center, the Netherlands. A patient list was generated with the following search criteria: "steatosis OR steatohepatitis OR NASH OR NAFLD" from 1997 onwards. Patient records were reviewed and patients diagnosed with HBV, HCV, alcohol-related steatohepatitis, and auto-immune hepatitis were excluded. In addition, patients with other comorbidities, treated with steroids, hormonal therapy

including anticonception and with benign liver tumors were excluded. Then, biobanked stained liver tissue slides were collected and reviewed by an external pathologist (JV). All patients who passed this review and were diagnosed as true MASLD patients were then included. Stored tissue blocks containing the biopsy sample were then requested from the tissue biobank and further processed for RNA isolation and next-generation sequencing. Stored serum samples collected at or close to the biopsy date (<1 year) were available for 29 of these patients with various grades of fibrosis (F0, $n = 4$; F1, $n = 2$; F2, $n = 10$; F3, $n = 9$; F4, $n = 4$). Due to the retrospective nature of this study, written informed consent was not obtained from each patient. Instead, the ethical review board of the Erasmus MC approved this study as it was in accordance with the COREON guidelines, which describes the use of coded-anonymous residual human tissue for scientific research (www.coreon.org).

## Transcriptome analysis

Next Generation Sequencing (NGS) was performed on frozen mouse liver tissue and human-archived formalin-fixed paraffin-embedded (FFPE) liver biopsies (using 10 μm tissue sections) from the translation cohort. Total RNA from the HFD-fed mice ($n = 12$) and the chow-fed mice ($n = 6$) was extracted using the Ambion RNAqueous total RNA isolation kit (Thermo Fisher Scientific, Inc, Waltman, MA, kit #AM1912). The RNA concentration was determined using a Nanodrop 1000 spectrophotometer (Isogen Life Science, De Meern, Netherlands), and RNA quality was evaluated using the 2100 Bioanalyzer (Agilent Technologies, Amstelveen, Netherlands). From the mouse liver RNA samples, strand-specific mRNA sequencing libraries for the Illumina platform (San Diego, CA) were generated and sequenced at BaseClear BV (Leiden, Netherlands). The libraries were multiplexed, clustered, and sequenced on an Illumina HiSeq 2500 using a single-read 50-cycle sequencing protocol, with 15 million reads per sample and indexing. Differentially expressed genes (DEGs) were identified using the DEseq2 method[44] with a statistical cut-off for false discovery rate of <0.001. The DEGs were used as input for pathway analysis through the Ingenuity Pathway Analysis (IPA) suite (www.ingenuity.com, accessed 2024).

From the human FFPE liver samples, a subset of 67 samples out of the 74 samples passed Quality Control (QC) and was selected for RNA sequencing. The QC process involved evaluation of the total RNA concentration and DV200 values, which represent the proportion of RNA fragments exceeding 200 nucleotides as determined from the outcomes of electropherograms[45]. From the selected samples, ribosomal RNA (rRNA) was depleted from total RNA (rRNA depletion kit NEB# E6310, Biolabs), the total RNA was processed into tagged random sequence libraries (NEBNext Ultra II Directional RNA Library Prep Kit, NEB #E7760S/L for Illumina, Biolabs) and sample quality was checked for proper size distribution (300–500 bp peak, Fragment Analyzer). The mixed (multiplex) sample libraries were sequenced on an Illumina NovaSeq6000 sequencer with a paired-read 150-cycle sequencing protocol at GenomeScan BV (Leiden, the Netherlands), resulting in at least 40 million reads per sample. The dataset of this study is accessible at the NCBI Gene Expression Omnibus (GEO) database with accession number GSE240729. DEGs across fibrosis stages were identified by DESeq2 ($p$-value < 0.01 and (2logR > 0.5 OR 2logR < −1) and avg(nCnts)>20. DEGs were used as the input for pathway and upstream regulator analysis using the Ingenuity Pathway Analysis (IPA) suite (www.ingenuity.com, accessed 2024).

## Candidate biomarker selection

Based on the combined gene expression results from the murine and human liver analyses, we identified a set of genes that correlated (using Spearman's rank and Pearson correlation analysis; Padj < 0.01) with newly synthesized collagens in the mice and were also expressed in liver tissue of patients with MASLD. Next, we determined which of these translatable candidate genes form proteins that can be measured in serum using ELISA assays. To this end, several steps were taken. First, candidate genes were selected that have a soluble protein as documented in the Cortellis Drug Discovery Intelligence database (Clarivate, https://www.cortellis.com). Next, proteins were selected for which ELISA assays were available. We evaluated whether these ELISA assays could measure the proteins in serum. Lastly, the accuracy of these assays was evaluated using spike-ins with protein from another supplier than the assay. From this selection process, 11 proteins that could be detected with the spike-in procedure and showed a good antibody recovery, were selected for analysis in serum samples from the cohorts.

## Selection of testing cohort

A testing cohort of 128 patients with MASLD from three academic centers in the Netherlands was selected to evaluate the accuracy of the chosen biomarker panel. Since this cohort is used for feature selection and the development of the machine learning model, the testing cohort should also be considered a discovery cohort. MASLD was defined as the presence of hepatic steatosis with at least one cardio-metabolic risk factor and no other causes of hepatic steatosis[2]. Patients with potential other causes of liver damage besides MASLD, including excessive alcohol use (women >14 units/week, men >21 units/week), were excluded. Of the 128 patients, 68 were enrolled in the Amsterdam MASLD-MASH cohort (ANCHOR) study[46]. Twenty-two participants were randomly selected from the group of patients with fibrosis stage F0 or F1 in the BARICO (Bariatric Surgery Rijnstate and Radboudumc Neuroimaging and Cognition in Obesity) study[47] to enrich the testing cohort with patients without fibrosis. The remaining 38 patients underwent liver biopsies as part of their clinical evaluation and staging process at the Leiden University Medical Center. All patients were above 18 years of age and provided written informed consent. Studies were approved by both the local Medical Ethics Committee and the local institutional ethics committee, and were conducted in compliance with the Declaration of Helsinki and according to Good Clinical Practice guidelines.

## Selection of independent validation cohort

A validation cohort was performed in a prospective cohort study of patients with MASLD (NCT04340817), which was approved by the Research Ethics Committee of the Capital Region of Denmark (H-17029039). Patients were recruited from January 2017 to December 2022 from the outpatient clinic at the Gastro Unit, Copenhagen University Hospital Hvidovre. The study conforms to the guidelines of the 1975 Declaration of Helsinki. All patients gave their informed consent to participate. After the collection of informed consent, patients with possible MASLD were assessed at a screening visit. In total, 156 patients were characterized based on their history, clinical assessments, blood tests (general screening including liver and metabolic tests), Fib-4, APRI, and FibroScan. Alcohol use was evaluated based on interviews combined with the Alcohol Use Disorders Identification Test-Concise (AUDIT-C) questionnaire. MASLD was defined as described above. If clinically significant fibrosis was suspected based on Fib-4 or FibroScan, patients were referred to a liver biopsy and a subsequent clinical visit where the patient was informed about the biopsy results.

## Liver histology assessment

Liver biopsy readings for both cohorts were performed in tandem by two specialized liver pathologists (JV and MD). The assessment of liver histology was conducted using the NAS scoring method and the MASLD scoring system devised by Kleiner and Brunt[48,49]. Steatosis was scored in grades 0-3. Grade 0 indicates that less than 5% of hepatocytes display lipid droplets, grade 1 corresponds to 5–33%, grade 2 represents 33–66%, and grade 3 denotes more than 66% of hepatocytes affected. Lobular inflammation is also scored on a scale of 0–3. Grade

0 signifies no inflammatory foci, grade 1 corresponds to less than 2 foci, grade 2 reflects 2–4 foci, and grade 3 indicates more than 4 inflammatory foci per 200× field. Hepatocyte ballooning is graded from 0 to 2. Grade 0 signifies no ballooned cells, grade 1 indicates few ballooned cells, and grade 2 represents the presence of many ballooned cells or prominent ballooning. To provide an overall assessment of NAFLD activity, the NAFLD activity score (NAS) was calculated as the sum of individual scores for steatosis, lobular inflammation, and hepatocyte ballooning. Lastly, fibrosis was assessed on a scale from stages F0–F4. Stage F0 denotes no fibrosis, stage F1 indicates perisinusoidal or periportal fibrosis, stage F2 reflects perisinusoidal and (peri)portal fibrosis, stage F3 signifies bridging fibrosis, and stage F4 represents cirrhosis.

## Machine learning analysis

We applied a Light Gradient Boosting Machine (LightGBM) model[38] to enable prediction into three subclasses: no fibrosis (F0/F1), mild fibrosis (F2) and advanced fibrosis (F3/F4). LightGBM is a machine learning technique, which is similar to random forests, and creates and combines an ensemble of decision trees. To train the model, the samples of the testing cohort were split into a balanced training set comprising of 30 samples each for each subclass, as well as a holdout test set consisting of 8 samples for no fibrosis, 8 samples for mild fibrosis, and 22 samples for advanced fibrosis. The feature importance is computed as previously described[50,51]. To compute feature selection using the LightGBM model, we first trained the model on the complete dataset and obtained the feature importance using the Genie index. Next, we selected the three most relevant features (SSc5D, Sema4D, and IGFBP7). Thereafter, we retrained the LightGBM model with the selected features and evaluated its performance to ensure the effectiveness of the feature selection. We performed linear discriminant analysis (LDA) on training data (limited to the testing cohort) and extracted two additional features (LD1 and LD2) to optimize the LightGBM model. Using the same LDA model we transformed the test data such that we had the same features for the test set. These features were provided together with the biomarkers to the LGBM model. The same LDA model as developed with the training data, was also used to predict fibrosis stages in the independent validation cohort. In order to assess the performance of the model, a 10-fold cross-validation approach was utilized. This involved dividing the training set further into 10 subsets, where the model was trained and evaluated using different combinations of these subsets as training and validation data. To evaluate the effect of adding clinical variables to the model, 20× cross validation was used. The accuracy and area under the receiver operating characteristic curve (AUROC) were calculated as performance metrics to evaluate the model's predictive capabilities. By employing this cross-validation strategy, the accuracy of the model was determined by comparing the predicted labels with the actual labels from the validation subsets. To evaluate the model, we used normalized serum protein levels from the independent validation cohort to assess accuracy, sensitivity and specificity. Normalization was performed by adjusting the mean of the validation set relative to the mean of the training set. Additionally, the AUROC metric was used to evaluate the model's ability to discriminate between different fibrosis stages, providing a measure of its overall performance.

## Statistical analyses

Continuous variables were expressed as means ± SD or SEM, or medians and inter-quartile ranges (IQR), categorical variables were expressed as numbers and percentages. Differences in clinical characteristics and fibrosis biomarkers between the fibrosis stages were tested by the unpaired Student's *t*-test and the Mann–Whitney *U*-test (for either normally or not normally distributed continuous 2 variables) or the chi-square test for categorical variables. Statistical tests are performed using two-sided statistics unless otherwise stated.

## Reporting summary

Further information on research design is available in the Nature Portfolio Reporting Summary linked to this article.

## Data availability

The transcriptome data generated in this study have been deposited in the NCBI Gene Expression Omnibus (GEO) database under accession code GSE240729. GEO Accession viewer (nih.gov). Source data are provided with this paper.

## Code availability

The codes used in this study are available via repositories https://codeberg.org/serdar-/TLM3-Fibrosis-Predictor.

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

## Acknowledgements

The in vivo work was generated by a consortium that was supported by grant 114025001 from ZonMW and the TNO research program "Predictive Health Technologies". The translation cohort work was supported by Health Holland, Top Sector Life Sciences & Health (LSH), grant number V201700972. In addition, clinical analyses were supported by a grant of the Rijnstate-Radboudumc Promotion Fund and the Shared Research Program GLoBAL, an initiative of Radboudumc, Rijnstate, and TNO supported by L.S.H. M.N. is supported by a personal ZONMW VICI grant 2020 [09150182010020]. A.G.H. is supported by the Amsterdam UMC Fellowship and TKI-PPP Health~Holland grants. A.B. was supported by: the European-Latin American ESCALON consortium, funded by the EU Horizon 2020 program, project number 825510, and the Foundation for Liver and Gastrointestinal Research (SLO). The authors express their gratitude to Miangela Lacle (Utrecht University Medical center, Utrecht, the Netherlands) for her insightful guidance in the selection of archived material. In addition, the authors thank the patients from the cohorts who contributed.

## Author contributions

R.H. and L.V. conceptualized the idea. R.H., L.V., and A.v.K. designed the experiments. L.V., R.H., and A.L.M. wrote the manuscript. J.S. and A.v.K. performed the animal experiment. M.P.M.C. analyzed and interpreted the omic data. J.S. and C.K. analyzed biomarker in serum. S.O. and S.D.

performed the computational modeling. D.v.d.M. performed the RNA-seq analysis in FFPE material. A.M.v.D., K.v.S., A.L.M., M.N., M.E.T., and A.G.H. collected samples from the testing cohort. E.B.R., P.N., M.P.W., and L.L.G. collected samples from the validation cohort. R.K., A.J.K., and E.J.H. collected samples from the testing cohort. A.B. and W.P.B. collected archived samples from the translation cohort. L.V., A.v.K., S.G., and R.H. performed biomarker selection. M.D. and J.V. performed pathological scoring of human MASLD biopsies. All authors discussed data and commented on the manuscript.

## Competing interests

The authors declare no competing interests.

## Ethical approval

TNO has a patent filed for the use of protein biomarkers for NAFLD (Inventors: L.V., A.v.K., and R.H.). Other authors declare no competing interests.

## Additional information

[1]TNO Healthy Living & Work, Leiden, The Netherlands. [2]Department of Vascular Medicine, Amsterdam University Medical Centers, Amsterdam, The Netherlands. [3]GenomeScan, Leiden, The Netherlands. [4]Gastro Unit, Copenhagen University Hospital Hvidovre, Hvidovre, Denmark and Department of Clinical Medicine, University of Copenhagen, Copenhagen, Denmark. [5]Department of Medical Imaging, Anatomy, and Radboud Alzheimer Center, Radboud University Medical Center, Donders Institute for Brain, Cognition, and Behavior, Nijmegen, the Netherlands. [6]Department of Bariatric Surgery, Vitalys, Rijnstate Hospital, Arnhem, the Netherlands and Division of Human Nutrition and Health, Wageningen University, Wageningen, The Netherlands. [7]Department of Gastroenterology and Hepatology, Erasmus MC University Medical Center, Rotterdam, The Netherlands. [8]Department of Pathology, Erasmus MC Cancer Institute, Rotterdam, The Netherlands. [9]Translational Medicine, Bristol Meyers Squibb, Princeton Pike, NJ, USA. [10]Good Biomarker Sciences, Leiden, The Netherlands. [11]Department of Pathology, Amsterdam University Medical Centers, Amsterdam, The Netherlands. [12]Department of Gastroenterology and Hepatology, Leiden University Medical Center, Leiden, The Netherlands. [13]These authors contributed equally: Lars Verschuren, Anne Linde Mak. [14]These authors jointly supervised this work: Adriaan G. Holleboom, Maarten E. Tushuizen, Roeland Hanemaaijer. ✉e-mail: lars.verschuren@tno.nl

