## [Peer Review File · Nature Communications]

Development of a novel non-invasive biomarker panel for hepatic fibrosis in MASLDEditorial Note: Parts of this Peer Review File have been redacted as indicated to remove third-party material where no permission to publish could be obtained.

REVIEWER COMMENTS

Reviewer #1 (Remarks to the Author):

In this manuscript entitled "Development of a novel non-invasive biomarker panel for hepatic fibrosis in MASLD," the authors explored novel fibrosis markers in MASLD. They used HFD-fed LDLr^{-/-}, Leiden mice as MASH mice model and measured gene expression levels in the liver. Besides, they also determined the fractional collagen synthesis rate using the D2O-labeling method, and extracted a group of genes that correlated strongly with the fractional collagen synthesis rate. On the other hand, using liver tissue from MASLD patients, they identified genes whose expression varied in cases of fibrosis development, narrowed down the candidate proteins that overlapped with the gene groups identified in mice, and further examined proteins that could be detected in blood, resulting in 11 candidate biomarkers. In the validation cohort, they confirmed that these candidate proteins fluctuate with fibrosis progression and that the combination of the three proteins can predict the stage of liver fibrosis well by machine learning.

MASLD is assumed to be the largest chronic liver disease, so it is clinically very important to find useful non-invasive fibrosis markers in MASLD. And use of D2O-labeling for marker search is a novel approach. However, as the authors also stated in the manuscript, there are no data on whether the combination of these three proteins is really clinically significant, so it is difficult to conclude that their marker panel is useful clinically. In addition, there are many points should be clarify as to why this method was taken in each step of the refinement of marker candidates.

1. The authors used the HFD-fed LDLr^{-/-}, Leiden mice, which exhibits the steatohepatitis phenotype. Why did the authors use the mice first? Is it not possible to use an alternative model such as Western diet for wild-type mice?
2. In this study, the fractional collagen synthesis rate by D2O labeling was measured in the

process of narrowing down the genes involved in fibrosis. However, it is unclear why D2O-labeling is chosen. It is well known that long-term HFD increases collagen production in the liver. Since the amount of collagen production in the liver correlates well with mRNA expression of fibrogenic genes such as Col1a1 and Col1a2, if we want to search for genes that increase with fibrosis progression, we can narrow down the genes that are correlated with Col1a1 and Col1a2.

3. MASLD is a new concept, and diagnostic criteria have been proposed by AASLD. This diagnostic criterion requires the presence of cardiometabolic risk factors, but there is no clinical information on these factors in this manuscript. Without this information, they cannot state that samples are from MASLD patients.

4. After identification of genes whose expression changed by fibrosis progression, they narrowed down the candidate proteins that overlapped with the gene groups identified in mice. As mentioned above, the degree of fibrosis progression correlates with the expression of fibrogenic genes. If they want to search for genes that fluctuate according to fibrosis progression, why not pick up genes that are strongly correlated with fibrogenic genes mRNA expression in human liver tissue? In fact, the expression levels of SSC5D, SEMA4D, and IGMP7 which were finally picked up as candidate proteins, are not that high in HFD-fed LDLr^{-/-} mice, giving the impression that the mice may not have been necessary to narrow down the markers.

5. When machine learning is used to determine the diagnostic performance of fibrosis progression, it is not possible to determine whether the extracted protein groups are useful or whether the machine learning has increased the diagnostic performance. If they would like to state that these marker proteins are clinically useful, how well the candidate proteins predict fibrosis compared to existing fibrosis markers without machine learning should be described.

6. In order to examine clinical utility, an appropriate cutoff value should be established for fibrosis diagnosis and validated in a separate cohort using the cutoff value to analyze positive predictive and false positive rates.

7. The main clinical problem in MASLD is not so much the development of fibrosis itself, but the development of cirrhosis, decompensated events, and liver cancer. If this group of marker proteins is clinically important, its association with the incidence of these events should also be examined.

Reviewer #2 (Remarks to the Author):

The authors attempt to establish a novel panel of blood-based biomarkers that is mechanistically linked to the fibrogenic process: transcriptomics + proteomics -> a nice approach of selection

General comments

-Manuscript well written and easy to read.

-This concept is not completely novel, as there have previously been studies using ML tools to integrate clinical and 'omics data such as Perakakis et al, <https://doi.org/10.1016/j.metabol.2019.154005>; Moolla et al, <https://doi.org/10.1111/apt.15710>; Castane et al, <https://doi.org/10.3390/biom11030473>. The authors should have referenced these studies, and placed theirs in the context of these studies. Some of these also examine fibrosis associated with MASH as outcome.

-It would also benefit from a head-to-head comparison with current non-invasive biomarkers

-Also, it would have been best to have an external clinical validation set

-Why was the LDLr-/- chosen as MASLD mouse model? It is not as reflective of human MASH, and was used here as the training set which in my mind is a bit problematic. By using the LDLR-/- model, they are likely to have missed out on truly biologically relevant biomarkers

-In the gene set enrichment analysis, F3+F4 groups were compared to F0. However, in the modelling, F3+F4 groups were compared to F0+F1. Based on Figure 4A, there are clearly DEGs between F1 vs. F0. It would be good to comment on this.

-It is unclear whether the fibrosis biomarker candidate list is based on the overlap of 645 DEGs chosen based on transcriptomics + proteomics in the mouse experiment (correlating with collagen synthesis) and the human liver transcriptomics data (Line 309-312)

-The 3 final biomarkers do not seem to be the most significant/changed biomarkers based

on Figure 4D-E. It would be good to highlight the 21 candidate blood-based biomarkers (line 323) or at least the 11 biomarkers that meet the criteria (328) on Figure 4D-E to show how significant their changes are and comment on this

-Although Sema4D was one of the three best performing features, it does not seem to change much in the serum of the independent validation cohort (supplementary figure 1), comment?

-Line 385-387: I do not see the direct correlation of all 3 biomarkers with dynamics of collagen deposition especially in the liver, except for IGFBP7 and Sema4D. Further, SSc5D is lowly expressed in the liver, comment?

Methods-associated Comments:

-How was the data normalized before presenting it to the LightGBM model?

-The full list of features used to train the models is not clearly specified. If both clinical and gene expression were used as features, how were they integrated?

- Fig. 5: - How were the features' importance computed?

- The ML model used 10-fold CV. How well do the feature importance hold up across the 10-folds?

- Fig. 6

- The confusion matrix reports the model's performance on the test set. However, this is a randomly selected subset of 38 samples and performance on this subset may not be indicative of the overall performance on the full 128 samples. Also, it is not clear why an unbalanced set of 8-8-22 samples were used in the test set.

- It would be good to provide additional performance metrics such as, NPV, PPV and F1-score (given that it is an unbalanced test set)

-Suppl Fig. 2

- It is very surprising that adding sex and BMI reduces the model's performance so significantly. Given that this is a decision tree-based ensemble model, shouldn't the model be able to pick the right features even when some additional features are provided?

- The fig. states that 20-fold CV was used whereas in the Methods section (Line 254) it is mentioned as 10-fold cross validation is used.

- Line 252: It is unclear how the LDA features were computed and used. Were they computed independently per fold? While the LDA features were deemed important they do not show up in Fig. 5. It is surprising the LightGBM model were unable to learn the features learnt using LDA. What was the performance without using LDA?

- Line 254: Given the 10-fold CV the validation set had 9 samples per fold, i.e., 3 samples per class per fold. Having such a small validation set can lead to the selection of a biased model. It would be good if the variability in performance across folds is reported. In general, in addition to providing the model's performance on the test set, it's performance on the training and validation set must also be reported.

Minor comments

- Suggestion: Since there are multiple sources of data (murine, human) and several methods of measurement (transcriptome, liver lipid analysis, proteomics etc.) as well as several analysis (DEG analysis, correlation analysis, ML-based modelling etc.), it would be good to have a block diagram that describes the connections between these.

-The AUC value for F0/F1 reported in Abstract Line 53 is 0.87 whereas the same value reported in Line 351 in the Results is 0.82.

-Line 247: The F0-F4 described in the ML analysis section are assumed to correspond to the Stage 0-4 of fibrosis described in Line 235.

-Figure 3c, 4c are not legible

-Inconsistency in writing the fibrosis stage

What are the triangles (inverted or not) depicting on Supplemental Figure 1?

-GSE data should be available, but did not receive the secure token

Dear reviewers,

We value your insights and constructive feedback. This gave us the opportunity to improve our manuscript. We are delighted to inform you that we have thoroughly addressed your comments as highlighted in this rebuttal. Our revised manuscript now includes additional results that not only address the your comments but also substantiate our previously reported findings.

REVIEWER COMMENTS

Reviewer #1:

In this manuscript entitled "Development of a novel non-invasive biomarker panel for hepatic fibrosis in MASLD," the authors explored novel fibrosis markers in MASLD. They used HFD-fed LDLr^{-/-}, Leiden mice as MASH mice model and measured gene expression levels in the liver. Besides, they also determined the fractional collagen synthesis rate using the D2O-labeling method, and extracted a group of genes that correlated strongly with the fractional collagen synthesis rate. On the other hand, using liver tissue from MASLD patients, they identified genes whose expression varied in cases of fibrosis development, narrowed down the candidate proteins that overlapped with the gene groups identified in mice, and further examined proteins that could be detected in blood, resulting in 11 candidate biomarkers. In the validation cohort, they confirmed that these candidate proteins fluctuate with fibrosis progression and that the combination of the three proteins can predict the stage of liver fibrosis well by machine learning.

MASLD is assumed to be the largest chronic liver disease, so it is clinically very important to find useful non-invasive fibrosis markers in MASLD. And use of D2O-labeling for marker search is a novel approach. However, as the authors also stated in the manuscript, there are no data on whether the combination of these three proteins is really clinically significant, so it is difficult to conclude that their marker panel is useful clinically. In addition, there are many points should be clarify as to why this method was taken in each step of the refinement of marker candidates.

1.1. . The authors used the HFD-fed LDLr^{-/-}, Leiden mice, which exhibits the steatohepatitis phenotype. Why did the authors use the mice first? Is it not possible to use an alternative model such as Western diet for wild-type mice?

We have selected this mouse model based on several years of studying and comparing different mouse models. The chosen model seemed to be comprehensive in terms of mimicking complex human pathophysiology and its translatability on multiple levels including histology, pathways, and plasma metabolome (e.g. van Koppen et al., Cell Mol Gastroenterol Hepatol 2017; Morrison et al., Hepatol Commun 2018; Morrison et al., Frontiers of Physiology 2018; Martinez-Arranz et al., Hepatology, 2022) as well as in its responses to exercise, nutritional and pharmaceutical treatments (e.g. van den Hoek et al., Metabolism, 2021; van den Hoek et al. Cells 2020 and unpublished data with a.o. selective agonists of thyroid hormone receptor- β).

Our choice for the LDLr^{-/-}.Leiden model was strengthened by a study published by Teufel et al. (Teufel et al. Gastroenterology. 2016), which compared various mouse models of MAFLD/MASH with human pathophysiology on the gene expression and pathway level. This study demonstrated minimal gene expression overlap when comparing liver tissues from patients representing different stages of the disease and nine MAFLD/MASH mouse models, including C57BL/6 mice on a Western-type diet. Using their open access datasets, we compared longitudinal gene expression datasets from various experimental models (including the HFD-fed LDLr^{-/-}.Leiden mouse) with the same human gene

dataset as Teufel et al. (GSE48452). This comparison revealed that the LDLr^{-/-}.Leiden model exhibits approximately a 50% similarity on the gene-level and 66% similarity on pathway level (see also van Koppen et al., Cell Mol Gastroenterol Hepatol. 2017), i.e. the highest overlap from all experimental models analyzed. Moreover, we have demonstrated that specific experimental conditions in the LDLr^{-/-}.Leiden model can recapitulate human disease at the molecular level, including disease pathways and upstream regulators (Morrison et al. Front. Physiol. 2018), and extensive head-to-head comparisons on the metabolome level with N=535 MAFLD/MASH patients (Morrison et al., Hepatol. Communic. 2018) and N=1099 patients (Martínez-Arranz et al., Hepatology 2022) further substantiated the translational character of experimental conditions in HFD-stimulated LDLr^{-/-}.Leiden mice. Because of these studies and other (longitudinal) studies in which we tested different diet-inducible MASH models including LDLr^{-/-}.Leiden mice, C57BL/6 mice, ob/ob mice and KKay mice (e.g. Abe et al. Biology Open 2019; van Koppen et al., Cell Mol Gastroenterol Hepatol 2017; Gart et al., Heliyon 2023), we were confident that the experimental model conditions being used in this study optimally reflect the MASLD and MASH-associated molecular pathways evoked in humans and would enable us to identify genes that correlate with collagen formation and that may serve as candidate biomarkers.

We followed the advice of the reviewer to include an alternative model based on wild-type mice, i.e. Western-type diet-fed C57BL/6 mice. We conducted a series of additional analyses to compare differential gene expression in LDLr^{-/-}.Leiden and C57BL/6 mouse models with the human gene expression relevant for liver fibrosis. To do so, we used livers from LDLr^{-/-}.Leiden mice (HFD vs. Chow) and, as suggested by the reviewer, livers from a new study with C57BL/6 mice fed a Western-type diet (WTD vs. Chow). The C57BL/6 mice developed hepatic fibrosis after 24 weeks on a WTD indicating successful disease induction. Using RNAseq analysis, we compared (head-to-head and using the same statistical criteria) the differential gene expression in these mouse models with that in human liver biopsies (i.e. patients characterized with biopsy-confirmed F4 fibrosis vs F0/1). Figure 1 in this rebuttal presents a proportional Venn diagram illustrating the overlap of gene expression between the HFD-fed LDLr^{-/-}.Leiden mice and WTD-fed C57BL/6 mice (all Padj<0.001), and patients with F4. The analysis demonstrates that mouse models only partially reflect the human gene expression profile and that the LDLr^{-/-}.Leiden mouse was more comprehensive and had greater overlap with humans and captured most of the genes identified in WTD-treated C57BL/6 mice. Only a small proportion of genes (30 genes) were uniquely found in WTD-treated C57BL/6 mice, most of which encoding for intracellular proteins that are not suitable as plasma biomarkers. By contrast, the HFD-fed LDLr^{-/-}.Leiden mice expressed much more human genes that were not identified in WTD-treated C57BL/6 mice (915 genes) which supports our initial choice to use the HFD-treated LDLr^{-/-}.Leiden mouse as a model. Thus the use of Western-type diet-fed C57BL6 mice instead of LDLr^{-/-}.Leiden mice would have resulted in a much lower set of candidate biomarkers.

We added a section in the discussion on the mouse model as well as a supplement figure 4 of the manuscript.

Figure 1. Venn diagram showing the overlap of DEGs in the Western-type diet-treated C57BL/6 mice and High fat diet-treated LDLr-/-Leiden mice as compared to respective chow fed controls. The DEGs of the mouse studies were compared to DEGs (all $P_{adj} < 0.001$) in human liver (comparison of biopsy confirmed F4 vs. F0/1 livers from MASLD patients).

1.2. In this study, the fractional collagen synthesis rate by D2O labeling was measured in the process of narrowing down the genes involved in fibrosis. However, it is unclear why D2O-labeling is chosen. It is well known that long-term HFD increases collagen production in the liver. Since the amount of collagen production in the liver correlates well with mRNA expression of fibrogenic genes such as Col1a1 and Col1a2, if we want to search for genes that increase with fibrosis progression, we can narrow down the genes that are correlated with Col1a1 and Col1a2.

It is correct that in some cases, the genes Col1a1 and Col1a2 exhibit a regulatory mRNA expression pattern that aligns with that of the translated protein. However, it has also been demonstrated that mRNA and protein concentrations can deviate because there are distinct regulatory mechanisms for these genes compared to their protein counterparts (Mia & Bank, Cell Tiss Res (2016); Schwarz, Biochem Biophys Rep (2015); Namba et al., Circulation (1997)). Our approach circumvents this issue and concentrates on D2O-labeled collagen protein and is thus independent of a potential mismatch between mRNA and protein or MAFLD/MASH-associated changes in Col1a1 and Col1a2 mRNA transcription or mRNA stability. In addition: collagen, known for its complexity, undergoes a range of post-translational modifications, including lysyl hydroxylation, prolyl hydroxylation, and crosslinking. These modifications significantly influence the final protein product and a stable deposition of collagen. Given this complexity, our research has focused on correlating biomarkers directly with deposited (newly synthesized) collagen. To understand the dynamics of collagen protein synthesis, we used D2O (heavy water) labeling. This method was a great tool for differentiating the newly formed collagen from the collagen already present in the tissue. While other dynamic markers could be used, D2O labeling was preferred due to its ease of administration in mice via drinking water, as opposed to more laborious methods needing daily injections.

1.3. MASLD is a new concept, and diagnostic criteria have been proposed by AASLD. This diagnostic criterion requires the presence of cardiometabolic risk factors, but there is no clinical

information on these factors in this manuscript. Without this information, they cannot state that samples are from MASLD patients.

The reviewer is right, these data were not included in the manuscript although the information was available to us. The patients in the testing cohort met the diagnostic criteria for MASLD as outlined by the AASLD (Ref. Rinella et al. Ann Hepatol 2024 Jan-Feb;29(1):101133). The distribution of patients was as follows: 8 patients met 1 criterion; 25 patients met 2 criteria; 33 patients met 3 criteria; 21 patients met 4 criteria; and 41 patients met all 5 criteria. In the material and methods section of the manuscript it now has been addressed that the patients included in the cohort are MASLD patients.

1.4. After identification of genes whose expression changed by fibrosis progression, they narrowed down the candidate proteins that overlapped with the gene groups identified in mice. As mentioned above, the degree of fibrosis progression correlates with the expression of fibrogenic genes. If they want to search for genes that fluctuate according to fibrosis progression, why not pick up genes that are strongly correlated with fibrogenic genes mRNA expression in human liver tissue? In fact, the expression levels of SSC5D, SEMA4D, and IGMP7 which were finally picked up as candidate proteins, are not that high in HFD-fed LDLr -/- mice, giving the impression that the mice may not have been necessary to narrow down the markers.

The reviewer is right that it would have been less laborious to identify potential biomarker genes by directly comparison to the expression of fibrotic genes in human liver tissue. This has been done by a number of other research groups. However, our focus was to identify the novel biomarkers based their relation to the dynamic process of new collagen formation. Since dynamic labeling studies are not easy to perform in humans, we chose an alternative route and conducted the dynamic labeling study in a relevant mouse model. As already indicated in the previous question of the reviewer, there's a noteworthy observation that Col1a1 and Col1a2 genes do not always exhibit the same regulatory patterns at the gene level as they do at the protein level.

1.5. When machine learning is used to determine the diagnostic performance of fibrosis progression, it is not possible to determine whether the extracted protein groups are useful or whether the machine learning has increased the diagnostic performance. If they would like to state that these marker proteins are clinically useful, how well the candidate proteins predict fibrosis compared to existing fibrosis markers without machine learning should be described.

Thank you for raising the interesting discussion on the role of machine learning (ML) in diagnosis. It is important to understand the diagnostic performance in fibrosis in relation to the levels of the protein biomarkers. As always, the diagnosis is depending on the biomarker levels in serum. A Machine Learning model uses these levels to define classifications; it can only increase the diagnostic performance if the biomarkers, individually or in combination, have sufficient predictive information about the fibrotic stage. In our view, a machine learning model is therefore eminently suitable to make predictions based on multi biomarker analyses.

To further show and validate the predictive value of the machine learning-based model, we conducted a second validation study in an independent patient cohort. This also gave us the opportunity to compare the performance of our model to that of other existing NITs. The study confirmed the power of the model (Figure 7 in the manuscript), the data regarding the NITs comparison are shown in figure 3 in this rebuttal and are added to the manuscript as supplemental figure 3. By providing these additional analyses, we aimed to demonstrate the clinical relevance of our findings.

1.6. In order to examine clinical utility, an appropriate cutoff value should be established for fibrosis diagnosis and validated in a separate cohort using the cutoff value to analyze positive predictive and false positive rates.

As commented above in our view the diagnostic value is not solely restricted to the absolute cutoff values of the biomarker proteins but is based on the intrinsic power of machine learning models to make decisions based on multiple biomarker concentrations. To underline the importance we have now validated the model generated from one study in a completely independent patient cohort, including the accuracy, specificity, sensitivity and prediction. These data are shown in Figure 7F and supplemental figure 3 of the manuscript.

1.7. The main clinical problem in MASLD is not so much the development of fibrosis itself, but the development of cirrhosis, decompensated events, and liver cancer. If this group of marker proteins is clinically important, its association with the incidence of these events should also be examined.

It is a known and valid point that the main clinical burden of MASLD is the presence of cirrhosis with decompensated events, and increased risk of liver cancer. However, all these complications start with fibrosis, or fibrosis increases the risk of complications. Fibrosis is therefore accepted as a proxy for metabolic dysfunction-associated liver end points. In this study we therefore first focused on the non-invasive diagnosis of liver fibrosis, which itself is a clinical challenge, but we agree completely with the reviewer that it is highly interesting and informative to also study the association of the biomarker set with the clinical end-points. We hope to perform this analysis in future studies.

Reviewer #2:

The authors attempt to establish a novel panel of blood-based biomarkers that is mechanistically linked to the fibrogenic process: transcriptomics + proteomics -> a nice approach of selection

General comments

-Manuscript well written and easy to read.

2.1 -This concept is not completely novel, as there have previously been studies using ML tools to integrate clinical and 'omics data such as Perakakis et al, <https://doi.org/10.1016/j.metabol.2019.154005>; Moolla et al, <https://doi.org/10.1111/apt.15710>; Castane et al, <https://doi.org/10.3390/biom11030473>. The authors should have referenced these studies, and placed theirs in the context of these studies. Some of these also examine fibrosis associated with MASH as outcome.

We thank the reviewer for the positive comments on the manuscript. The reviewer is right that other studies have been performed that use ML to integrate clinical and omics data. However, as far as we know these studies are based on combining omics data with pathological phenotypes such as fibrosis associated with MASH. The novel approach of our study is that we include dynamics of the pathological process in the identification of new biomarkers. So directly relate the novel biomarkers to the process of fibrosis formation. To address that ML models have been used before for the identification of new biomarkers we have referred to the Castane et al review- paper in the discussion section.

2.2 It would also benefit from a head-to-head comparison with current non-invasive biomarkers

We agree with the reviewer that comparing our novel biomarker-based model, hereafter referred to as the TLM3 model (TNO LGBM MASLD model, based on three biomarkers) with existing non-invasive tests (NITs) could yield significant insights. To perform a relevant head-to-head comparison all NITs

should be analyzed in one identical cohort. Therefore, we initiated a new collaboration with professor in hepatology, Prof. Dr. Lise Lotte Gluud from the Copenhagen University and Hospital Hvidovre in Denmark. She provided us with 156 serum samples and non-invasive biomarker data from their prospective cohort study of patients with biopsy-proven MASLD and fibrosis (NCT04340817). We have used the data from this independent cohort to compare the performance of our novel model with three established NITs, i.e. FIB-4, APRI and FibroScan. The patients were divided into 3 groups F0/F1 (66 patients), F2 (30 patients) and F3/F4 (60 patients). The data obtained from this independent validation study has been added to the manuscript (Results section and Figure 7) and a description of the clinical characteristics of the patients in the validation cohort has been added to Table 3 of the manuscript and the Material and Methods section.

We first assessed the accuracy of our TLM3 model in predicting all Fibrosis score subgroups (see also response to reviewer's next question 2.3) and compared it with the aforementioned non-invasive tests. Furthermore, a detailed head-to-head comparison was conducted focusing on the Sensitivity, Specificity, and Precision of each NIT and their predictive value.

Our findings indicate that in the independent validation cohort from Denmark, our model, demonstrates superior accuracy over the other NITs, FIB4, FibroScan and APRI (Figure 3F in this rebuttal; Figure 7F in the manuscript). In terms of Sensitivity, Specificity, and Precision, TLM3 proved to be more robust compared to its counterparts. All NITs, including TLM3 are less predictable for the F2 subgroup versus the other subgroups. Among the tests, FIB4 and APRI show effectiveness in predicting F0/F1 stages, while FibroScan demonstrates the ability in predicting F3/F4 stages. However, FIB4 and APRI are found to be poor predictors for F3/F4 stages, and FibroScan is not effective in predicting F0/F1 stages. TLM3 stands out for its ability to accurately predict both F0/F1 and F3/F4 stages, as reflected in its highest overall accuracy. (Figure 2 in this rebuttal; Supplemental Figure 3 in the manuscript).

Figure 2: Overview of model performances of different NITs in Sensitivity, Specificity, and Precision of A) Fibrosis stages F0/F1, B) Fibrosis stage F2, and C) Fibrosis stages F3/F4.

2.3 – Also, it would have been best to have an external clinical validation set

We agree with the reviewer that an additional independent cohort for predicting biomarker performance would be very informative. As mentioned above, we initiated a collaboration with Prof. Dr. Lise Lotte Gluud (MD) from the Copenhagen University and Hospital Hvidovre in Denmark. The patients of this additional validation cohort were divided into 3 groups F0/F1 (66 patients), F2 (30 patients) and F3/F4 (60 patients). The data obtained from this independent validation study has been added to the manuscript (results section and Figure 7) and a description of the clinical characteristics of the patients in the validation cohort has been added to the Material and Methods section.

We analyzed serum protein levels for the identified potential biomarkers in an independent Danish prospective MASLD cohort, in a similar way as performed in the testing cohort. Following these measurements, we applied the TLM3 model on the protein data to predict the F-stage of these patients. The new data confirmed the predictive accuracy of our initial study, with an AUC of 0.84 for both the F0/F1 and F3/F4 subgroups (see Figure 3 in this rebuttal; Figure 7 in the manuscript). The prediction of the F2 subgroup was more modest in this Danish external validation cohort than in the Dutch cohorts from the initial study.

Figure 3: Serum levels of A) IGFBP7, B) SSc5D, C) SEMA4D as measured in samples from patients from the Danish validation cohort. Values represent mean \pm SD. *indicates $P < 0.05$. D) AUROC curve to show the predictive value of this set of biomarkers to distinguish the individual MASLD fibrosis F-scores (F0/F1 versus F2 versus F3/F4). E) Confusion matrix of the hold-out set of predicted and true classes. F) Overview of model performances (accuracy) of different NITs in predicting fibrosis.

2.4 Why was the LDLr^{-/-} chosen as MASLD mouse model? It is not as reflective of human MASH, and was used here as the training set which in my mind is a bit problematic. By using the LDLR^{-/-} model, they are likely to have missed out on truly biologically relevant biomarkers.

We understand this concern because the identification of our biomarkers is primarily based on gene expression changes and deuterated water labeling analyses in high fat diet-treated fed LDLr^{-/-}.Leiden mice developing MAFLD and fibrosis. We address this concern and checked for potentially missed biomarkers by a head-to-head comparison of a Western diet fed mouse model (C57BL/6+WTD) and HFD-fed LDLr^{-/-}.Leiden mouse (fed LDLr^{-/-}.Leiden) using human gene expression profiling as ground truth to estimate the overlap of each model with human gene expression changes. See also data in figure 1 in this rebuttal and supplemental figure 4 in the manuscript.

Figure 1 in this rebuttal demonstrates that the chosen experimental conditions (HFD-fed LDLr^{-/-}.Leiden mice) reflects the human gene expression profile that is characteristic for F4 patients better than a WTD-treated C57BL/6 mouse model which is a conventional MAFLD/MASH model. The new comparative data also demonstrate that the chosen model expresses many (915) human genes that are not expressed in the C57BL/6 mouse, while only a few (30 genes) were uniquely expressed by HFD-treated C57BL/6 mice. The vast majority of these 30 genes encoded for intracellular proteins, such as transcription factors (NR1I3, GLIS2), membrane proteins (CXCR4, GPNMB, PRLR, SDCBP2) and intracellular kinases (CKB, MAPK15, PPM1K), excluding the potential use as plasma biomarkers. In contrast to C57BL/6, a large proportion of DEGs in HFD-fed LDLr^{-/-}.Leiden mice were unique (915 genes) and overlapped with human DEGs and were not affected in the WTD-fed C57BL/6 mice. Among these genes were 3 (SSc5D, FBN and PLAU) out of 11 of the biomarkers described herein. These data demonstrate that the use of WTD-fed C57BL/6 mice instead of HFD-treated LDLr^{-/-}.Leiden mice would have resulted in a much lower set of overlapping DEGs with humans, and would have resulted in missing at least 3 of the described biomarkers. Altogether, we believe that our initial decision to use the HFD-fed LDLr^{-/-}.Leiden mouse for this study was the best possible because it was selected on data indicating high translatability to human disease which was confirmed by the new results from WTD-fed C57BL/6 mice. For details regarding the choice of the mouse model please see our response to question 1.1 of reviewer 1.

Since we had, in our view, a relevant mouse model for MASLD we used this as a starting point for novel biomarker identification. The advantage of this mouse model is that it enabled us to use the dynamics of fibrogenesis for the identification of novel biomarkers which, for obvious reasons, is not possible to do in a human study.

2.5 In the gene set enrichment analysis, F3+F4 groups were compared to F0. However, in the modelling, F3+F4 groups were compared to F0+F1. Based on Figure 4A, there are clearly DEGs between F1 vs. F0. It would be good to comment on this.

The reviewer is right, it seems to be inconsequent to use different comparisons in the gene set enrichment analysis and the modeling part of the biomarkers. We decided to use F0 as a control to include all potential biomarkers and prevent missing out on relevant ones. However, since pathologists explained that in clinical practice it is difficult to discriminate between F0 and F1 fibrosis, we chose to use F0/F1 for the modeling phase to make the model more robust and relevant.

To analyze the potential effect of these choices, we also performed the gene set enrichment analysis compared to F0/F1 instead of F0 (See Table 1 below). This analysis showed that as compared to F0, the comparison to F0/F1 did not have consequences for the significance of the biomarkers in human biopsies.

		F3/F0	F4/F0	F3/F01	F4/F01
	Entrez Gene Name	pval	pval	pval	pval
IGFBP7	insulin like growth factor binding protein 7	3,1E-06	4,6E-13	6,3E-08	3,6E-16
SSC5D	scavenger receptor cysteine rich family member with 5 domains	1,9E-07	6,0E-09	3,9E-11	2,5E-12
SEMA4D	semaphorin 4D	2,5E-04	2,5E-08	3,8E-09	3,6E-14

Table 1. Statistical significance of biomarkers on gene level if compared vs F0 or as compared to F0/F1.

2.6 -It is unclear whether the fibrosis biomarker candidate list is based on the overlap of 645 DEGs chosen based on transcriptomics + proteomics in the mouse experiment (correlating with collagen synthesis) and the human liver transcriptomics data (Line 309-312)

We apologize for not being clear. The reviewer is right that we have performed correlation analysis of the mouse transcriptome with the mouse proteome to select the 645 signature genes. This was followed by an overlap analysis of these genes with the DEGs from human liver tissue samples (F3 and F4 relative to F0). See figure 4D and E in the manuscript. We have adapted the text in the manuscript to clarify this procedure.

2.7 The 3 final biomarkers do not seem to be the most significant/changed biomarkers based on Figure 4D-E. It would be good to highlight the 21 candidate blood-based biomarkers (line 323) or at least the 11 biomarkers that meet the criteria (328) on Figure 4D-E to show how significant their changes are and comment on this

It is correct that for the selected biomarkers at gene level (fold induction) they are not the most significantly changed ones. However, gene expression and protein expression levels are not always directly correlated. We therefore chose to do the selection based on significance (p-value) rather than fold induction. This resulted in a human relevant blood based biomarker set which is not one-to-one reflected by the strongest regulation at gene level in mouse. Figures 4D-E have been adjusted according to the reviewer's suggestion; the 11 biomarkers that met the criteria are marked in red, for clarity reasons we kept the marking of three biomarker names instead of marking all 11 biomarkers.

2.8 Although Sema4D was one of the three best performing features, it does not seem to change much in the serum of the independent validation cohort (supplementary figure 1), comment?

We are sorry for the confusion due to having used various cohorts for different purposes. The first selection of biomarkers was based on a preclinical study in which genes were identified that were related to new collagen deposition in the liver. A next step in the selection procedure was to check whether the identified mouse genes were also present in human, and whether these genes were translated in soluble secreted proteins which could be detected in the circulation. The following step was whether commercial ELISAs were available and to validate whether these ELISAs were applicable in human serum samples. The data in Supplemental Figure 1 was used as a qualitative check whether the resulting selection of biomarkers could be measured in human serum of MASH patients. The number of samples was very low. For clarity we have now added the number of samples in the figure legend. The conclusion of this analysis was that finally 11 biomarkers passed the selection for further analysis in the testing cohort (128 samples). Sema4D was one of these 11 selected biomarkers. It showed upon analysis, including ML modelling, that together with IGFBP7 and SSC5D, Sema4D was in the top 3 of the feature selection, as shown in Figure 5.

We have adapted the text in the manuscript to prevent the observed confusion.

2.9 -Line 385-387: I do not see the direct correlation of all 3 biomarkers with dynamics of collagen deposition especially in the liver, except for IGFBP7 and Sema4D. Further, SSc5D is lowly expressed in the liver, comment?

The correlation of the gene expression of the three biomarkers with the dynamics of collagen deposition has been shown in Supplemental Table 1. The reviewer is right that the gene expression level for SSc5D is low in mouse liver (column F), however, all three genes are significantly upregulated in HFD fed LDLr^{-/-}.Leiden mice as compared to control chow (columns G and H). Moreover, the three genes are directly and strongly correlated with the collagen protein deposition in the liver tissue (columns I-M), showing correlation coefficients between 0.86 and 0.98. This result led to taking also these three biomarkers (as well as others) as potential candidates to further selection whether they are regulated in human and present in the circulation. The study described in this manuscript shows that all three biomarkers are well detectable in the circulation of MASH patients and correlated to the development of fibrosis.

Methods-associated Comments:

2.10 How was the data normalized before presenting it to the LightGBM model?

The dataset was not subjected to normalization (scaling) prior to its application in the LightGBM algorithm. It is important to note that, in contrast to linear regression models, decision tree-based algorithms—such as Random Forest and Gradient Boosting Machines—do not necessitate data normalization. This is because their operational focus is on the segmentation of data rather than its variance. Consequently, normalization of the data would merely alter the threshold values utilized by the decision trees within the algorithm, without significantly impacting the model's performance.

2.11 -The full list of features used to train the models is not clearly specified. If both clinical and gene expression were used as features, how were they integrated?

The features included to train the model were the 3 biomarkers selected from the discovery cohort, viz. "IGFBP7 (ng/ml)", "SSC5D (ng/ml)", "SEMA4D (ug/ml)", along with two additional engineered features, LD1 and LD2. These latter features were derived through a transformative process utilizing a Linear Discriminant Analysis (LDA) model. The generation of LD1 and LD2 involved applying the LDA model to the aforementioned biomarkers - "IGFBP7 (ng/ml)", "SSC5D (ng/ml)", and "SEMA4D (ug/ml)". Initially, we fitted the training data into the LDA model. This model is designed to learn a transformation that maximizes the variance between classes (in our case, the classes refer to different fibrosis stage groups) while minimizing the variance within each class. The result of this transformation is a dataset with reduced dimensionality, specifically two dimensions, following the principle that the number of dimensions is N-1 for N classes.

Subsequently, the transformed dataset, now encompassing these two new dimensions (LD1 and LD2) along with the original biomarker features, was utilized as a five-feature input for the LightGBM model. This method was previously performed in a paper described by Chen et al. *Sensors* 2019, 19(7), 1631. This approach allowed us to leverage the enhanced discriminative power of the LDA-transformed features in conjunction with the original biomarkers to improve the predictive performance of our LightGBM model in the context of MASLD.

2.12- Fig. 5: - How were the features' importance computed?

The feature importance is computed as described (ref. Ke et al. Adv Neural Inf Process Syst. 2017;2017-Decem(Nips):3147–55 and Ayyanar et al. 6th Int. Conf. Trends Electr. Informatics (ICOEI). 2022. p. 117–22). To compute feature selection using the LightGBM model, we first trained the model on the complete dataset and obtained the feature importance using the Genie index. Next, we selected the three most relevant features (SSc5D, Sema4D and IGFBP7) by choosing the top 3 features. Thereafter we retrained the LightGBM model with the selected features and evaluated its performance to ensure the effectiveness of the feature selection. We have added the feature selection procedure in the material and methods section.

2.13 - The ML model used 10-fold CV. How well do the feature importance hold up across the 10-folds?

In Figure 5, we have illustrated the distribution of feature importances using box plots to represent how the features hold up across 10 folds. Each box plot shows the variance in importance scores for a particular feature throughout the cross-validation folds.

2.14 - The confusion matrix reports the model's performance on the test set. However, this is a randomly selected subset of 38 samples and performance on this subset may not be indicative of the overall performance on the full 128 samples. Also, it is not clear why an unbalanced set of 8-8-22 samples were used in the test set.

The reviewer is right that the test set was relatively small and unbalanced. This is due to the fact that in the training set we needed to use a balanced dataset (30-30-30) to train the model and to avoid bias. All remaining 38 samples of this cohort were included in the test set. The consequence of using a balanced training set is that the test set was unbalanced. Therefore, also in response to the reviewers, we used an additional independent validation cohort of 156 patients with biopsy proven MASH and fibrosis and performed analyses of the three biomarkers. The performance of the model in predicting fibrosis in this cohort is reported in this rebuttal and added to the manuscript (Figure 7).

2.15 It would be good to provide additional performance metrics such as, NPV, PPV and F1-score (given that it is an unbalanced test set)

Additional performance metrics such as accuracy, specificity and sensitivity, and prediction have been provided for the analysis in the second validation cohort because this dataset is larger (n=156 patients) and completely independent. These data have been added to the text in the results section, in figure 7F and in supplemental figure 3 in the manuscript.

2.16 - It is very surprising that adding sex and BMI reduces the model's performance so significantly. Given that this is a decision tree-based ensemble model, shouldn't the model be able to pick the right features even when some additional features are provided?

The reviewer is right that it is surprising that some additional biomarkers reduces the model's performance. On the other hand, it has been described before (e.g. Pfeifer arXiv: 2306.03702 (2023); Chen et al., Chemometrics Intell Lab Systems 192, 54-64 (2019); Wang et al., Infrared Physics Technol 123, 104191(2022)) that these ML models have a tendency to overfit noisy or irrelevant features, which can result in decreased general performance. LGBM is a random forest model which suffers from similar issues in handling noisy and/or irrelevant features. So if the clinical variables should have contributed to the prediction, it would have been resulted in better performance outcome. Our results, even after 20x cross-validation, showed no better performance, concluding that the clinical variables such as sex and BMI did not contribute to a better model.

2.17- The fig. states that 20-fold CV was used whereas in the Methods section (Line 254) it is mentioned as 10-fold cross validation is used.

The reviewer is correct, we used 10-fold CV for the LGBM model performance of the biomarkers. For the evaluation of the clinical variables to the model, 20x cross validation was used. We have adjusted this in the methods section accordingly.

2.18 - Line 252: It is unclear how the LDA features were computed and used. Were they computed independently per fold? While the LDA features were deemed important they do not show up in Fig. 5. It is surprising the LightGBM model were unable to learn the features learnt using LDA. What was the performance without using LDA?

The reviewer is right that we did not explain the feature engineering of the method explained completely. Therefore we elaborate on this topic here and adjusted the Method section accordingly. We trained an LDA model based on the training data. Then, we transformed the training data using the LDA model. This resulted in 2 extra features LD1 and LD2. Using the same LDA model we transformed the test data such that we had the same features for the test set. These features were provided together with the biomarkers to the LGBM model. The same LDA model as developed with the training data, was also used in the same way to make prediction from the biomarker data in the Danish cohort. They were not computed per fold, but calculated based on the whole training set. Figure 5 of the manuscript was used to determine the best performing biomarkers, the LD-features were not taken into account in this feature selection.

We used the LDA to optimize the final model resulting in a somewhat better performance. Without the LDA, overall performance was AUC 0.833 while adding the LD1 and LD2 the performance of the trainings set improved to AUC 0.87. Accuracy of the LDA alone on trainings set is 0.68. This model was used in the validation cohort with the reported performances as described in this rebuttal and the manuscript.

2.19- Line 254: Given the 10-fold CV the validation set had 9 samples per fold, i.e., 3 samples per class per fold. Having such a small validation set can lead to the selection of a biased model. It would be good if the variability in performance across folds is reported. In general, in addition to providing the model's performance on the test set, it's performance on the training and validation set must also be reported.

As indicated above, we agree with the reviewer that the validation set of the testing cohort is rather small. Therefore, as suggested by the reviewer, we added the performance throughout the folds. The training model has an overall AUC of 0.87 +/- 0.083 based on all subgroups and its overall accuracy is 0.744 +/- 0.071. These data have been added in the text of the results section. The deviation of the accuracy is not large among the folds in the CV and therefore the model performance could be indicated as sufficient. The best validation of the model is to use an additional independent validation cohort. Therefore, 156 patients were included from an additional Danish cohort. The performance of our model was evaluated in this independent validation cohort which is reported in this rebuttal (figure 2) and in the manuscript (Figure 7).

Minor comments:

2.20 Suggestion: Since there are multiple sources of data (murine, human) and several methods of measurement (transcriptome, liver lipid analysis, proteomics etc.) as well as several analysis (DEG analysis, correlation analysis, ML-based modelling etc.), it would be good to have a block diagram that describes the connections between these.

For clarification we generated an overview of the steps taken in the project to highlight multiple sources of data and methods of measurements. If the reviewer and editor like this graphical abstract, we can add this as a figure in the supplements.

[**editorial note:** figure redacted]

2.21 The AUC value for F0/F1 reported in Abstract Line 53 is 0.87 whereas the same value reported in Line 351 in the Results is 0.82.

Thank you for observing this important typo in the abstract. We have adjusted the abstract.

2.22 Line 247: The F0-F4 described in the ML analysis section are assumed to correspond to the Stage 0-4 of fibrosis described in Line 235.

The reviewer is correct, we have adjusted the wording in the Methods section and also checked for consequent description of the fibrosis stage throughout the manuscript.

2.23 Figure 3c, 4c are not legible

We have adjusted the figures 3c and 4c to enable readability.

2.24 Inconsistency in writing the fibrosis stage

The reviewer is correct; we have adjusted the wording throughout the text to make it more consequent

2.25 What are the triangles (inverted or not) depicting on Supplemental Figure 1?

The reviewer is correct that Supplemental Figure 1 was not clear, we have adjusted the legend of Suppl Fig 1 to clarify the triangles in the figures.

2.26 GSE data should be available, but did not receive the secure token

For GSE240729, please find here the secure token to access the dataset: : **cxqpoaakhrexbmt**

REVIEWER COMMENTS

Reviewer #1 (Remarks to the Author):

In 1.2, they responded that the reason D2O labeling was necessary was because mRNA and protein concentrations can deviate because there are distinct regulatory mechanisms for these genes compared to their protein counterparts. I understand the authors' contention that D2O labeling is a more appropriate model to evaluate the dynamics of collagen protein synthesis. The important question, however, is whether D2O labeling is useful in narrowing down biomarkers for MASLD. The authors have shown neither previous reports nor any data showing that using D2O labeling to narrow down candidate proteins is more useful than looking at correlations with col1a1 mRNA. It is necessary to show how much difference there is in the content of candidate proteins between the refinement by D2O labeling and the refinement by correlation with Col1a1 mRNA. In addition, the usefulness of D2O labeling cannot be claimed unless the final extracted protein is one that can be picked up by D2O labeling but not by correlation with Col1a1 mRNA.

In 1.4, the authors replied that the reason they used mice to narrow down the candidate proteins was because D2O labeling was necessary. As noted above, there is concern as to whether D2O labeling was necessary in this study. Are the 11 proteins that ultimately remained and the 3 proteins used as biomarkers not correlated with Col1a1 mRNA or Col1a2 mRNA in human liver tissue? If they strongly correlate with Col1a1 mRNA or Col1a2 mRNA, then this study would not be much different from the results of biomarker searches without D2O labeling experiments using mice, which would reduce its importance.

In 1.5 and 1.6, I suggested that the final three remaining proteins (IGFBP7, SSc5D, SEMA4D) should be described in terms of their ability to diagnose fibrosis on their own (not combined), including cutoff values. However, the authors did not answer this question. There are many papers out there that use machine learning to improve the diagnostic performance of diseases using existing parameters. If the authors would like to argue that they have found a new biomarker by narrowing down the parameters using D2O labeling in this study, they must have sufficient fibrosis diagnostic power without machine learning. In addition, the validation set is small, with only 156 cases, and particularly, there are only

30 cases of F2. While the validation results are crucial to demonstrating the usefulness of these findings, unfortunately, the number is not sufficient.

In reply 1.7, the authors argued that fibrosis was an acceptable endpoint. Since there have been many papers claiming to have discovered novel fibrosis markers, a biomarker that predicts only fibrosis is not novel. If the authors do not know whether a novel fibrosis marker has the ability to predict liver-related events including decompensation events and cancers, then an extremely high level of fibrosis diagnostic ability should be required. At each stage, the diagnostic performance must be statistically significantly higher than that of existing fibrosis markers, such as FIB-4 index.

Reviewer #2 (Remarks to the Author):

Thank you for addressing the concerns raised; this has significantly improved the overall consistency of the manuscript. However, there are still some issues that need to be addressed as follows:

For 2.10

2.10 How was the data normalized before presenting it to the LightGBM model?

Reviewer Response: While it is partially true that the model's results on the dataset used in this study might not be significantly altered by scaling/normalizing the data, the purpose of data scaling is to ensure that the methods developed here can be extended to many other datasets. These external datasets might have different data distributions, due to even minor differences in measurement techniques. The thresholds learnt by the proposed model may not produce the reported performance in such cases. One of the objectives of developing an ML model is to show that it has generalizability and data normalization is essential for that. Moreover, the results presented here need to consider the effect of this normalization. For instance, if a feature is normalized to a standard normal distribution, the parameters for this transformation are estimated from the training set (independently per fold). The validation/test sets are transformed using the learnt parameters before inference. This provides a realistic estimate of how the model is likely to generalize to other datasets.

2.11 -The full list of features used to train the models is not clearly specified. If both clinical and gene expression were used as features, how were they integrated?

Reviewer Response: Thanks for the clarification. However, please address the concern raised in 2.18 regarding the LDA computations.

2.16 - It is very surprising that adding sex and BMI reduces the model's performance so significantly. Given that this is a decision tree-based ensemble model, shouldn't the model be able to pick the right features even when some additional features are provided?

Reviewer Response: This necessitates the need to show if the model is biased towards a particular gender. This can be clarified by reporting the model's performance stratified by gender.

2.18 - Line 252: It is unclear how the LDA features were computed and used. Were they computed independently per fold? While the LDA features were deemed important they do not show up in Fig. 5. It is surprising the LightGBM model were unable to learn the features learnt using LDA. What was the performance without using LDA?

Reviewer Response: Thanks for the clarifications. In this case, the 10-fold CV results are biased due to leaky pre-processing, as the entire training set, including the validation sets used in CV, was used to compute the LDA features. This is akin to including the validation/test set to select (learn) features and then evaluate the feature's performance on the validation/test set.

2.20 Suggestion: Since there are multiple sources of data (murine, human) and several methods of measurement (transcriptome, liver lipid analysis, proteomics etc.) as well as several analysis (DEG analysis, correlation analysis, ML-based modelling etc.), it would be good to have a block diagram that describes the connections between these.

For clarification we generated an overview of the steps taken in the project to highlight multiple sources of data and methods of measurements. If the reviewer and editor like this graphical abstract, we can add this as a figure in the supplements.

Reviewer Response: Yes, please add the figure as it enhances clarity in the manuscript.

Dear reviewers (resubmitted 29MRT2024),

Thank you for your additional feedback and thoughts on our replies in the previous rebuttal. We recognize that some of our previous answers may not have been as clear as intended. In this revised rebuttal, we provide more detailed clarifications and have incorporated additional analyses to strengthen our manuscript. These again substantiate the final manuscript.

Reviewer #1 (Remarks to the Author):

“In 1.2, they responded that the reason D2O labeling was necessary was because mRNA and protein concentrations can deviate because there are distinct regulatory mechanisms for these genes compared to their protein counterparts. I understand the authors' contention that D2O labeling is a more appropriate model to evaluate the dynamics of collagen protein synthesis. The important question, however, is whether D2O labeling is useful in narrowing down biomarkers for MASLD. The authors have shown neither previous reports nor any data showing that using D2O labeling to narrow down candidate proteins is more useful than looking at correlations with col1a1 mRNA. It is necessary to show how much difference there is in the content of candidate proteins between the refinement by D2O labeling and the refinement by correlation with Col1a1 mRNA. In addition, the usefulness of D2O labeling cannot be claimed unless the final extracted protein is one that can be picked up by D2O labeling but not by correlation with Col1a1 mRNA.”

Author response: We agree that there are many valid ways to come to a set of candidate biomarkers. Correlating COL1A1 mRNA with gene transcripts of the candidate biomarkers could be one of them. It is a fairly simple and straight-forward approach and it is very likely that it has been tested by others, but did apparently not lead to the same biomarkers reported by us herein. Our biomarkers are novel and were identified using de novo formed COL1A1 protein as a primary metric for the correlation analysis.

To address the reviewer's request for a direct comparison, we performed additional analyses to compare the usefulness selecting candidate biomarkers based on COL1A1 mRNA expression versus D2O labeling.

- 1) Our additional analysis revealed that when using D2O labeling for correlation analysis, we identified 645 genes with significant correlation ($R^2 > 0.9$, as detailed in the manuscript). In contrast, correlating with Col1a1 mRNA instead of D2O labeling resulted in 434 correlated genes ($R^2 > 0.9$).*
- 2) If we zoom into the key biomarkers of our manuscript (Table 1), 5 out of 11 would also have been selected using COL1A1 mRNA for comparison (see Table 1, SSc5D, TNC, PLAU, VCAN and FBN1). However, among the biomarkers identified, 6 out of 11 biomarkers would **not** have been identified using COL1A1 mRNA correlation (see bold and italic selection in Table1). Crucially, 2 of our final 3 biomarkers (SEMA4D and IGFBP7) would **not** have been picked up using COL1A1 mRNA correlation.*

These results underscore the added benefit of D2O labeling for biomarker selection in our study, indicating D2O labeling correlation analysis provides a significant advantage over COL1A1 mRNA correlation in our study, making it essential and more effective for the selection of relevant biomarkers.

	GeneExpr
Symbol	COL1A1
IGFBP7	0.89
SSC5D	0.99
SEMA4D	0.87
TNC	0.96
PLAU	0.96
CXCL10	0.78
THBS1	0.82
PAM	0.86
VCAN	0.96
ADAMTS2	0.86
FBN1	0.95

Table 1. Correlation coefficient of genes in the mouse study related to the expression of COL1A1. Genes in bold and italic would **not** have been selected by correlation analysis using COL1A1 mRNA expression.

“In 1.4, the authors replied that the reason they used mice to narrow down the candidate proteins was because D2O labeling was necessary. As noted above, there is concern as to whether D2O labeling was necessary in this study. Are the 11 proteins that ultimately remained and the 3 proteins used as biomarkers not correlated with Col1a1 mRNA or Col1a2 mRNA in human liver tissue? If they strongly correlate with Col1a1 mRNA or Col1a2 mRNA, then this study would not be much different from the results of biomarker searches without D2O labeling experiments using mice, which would reduce its importance”.

Author response: As noted above, the mouse study and the selection using D2O labeling was essential to select the final biomarker candidates.

To address the reviewer's request we also performed a correlation analysis on the expression of the 11 proteins with human COL1A1 and COL1A2 expression in patient samples. The results are presented below in Table 2. This analysis indicates only moderate correlations between the gene expression of the identified biomarkers and both COL1A1 and COL1A2 expression in human liver tissue. Please compare this with the D2O correlations where we applied correlation cut-off >0.9, see manuscript. These data illustrate that compete current candidate biomarker selection would not have been selected if we would have used correlation with COL1A1 and COL1A2 expression.

We trust that this expanded analysis and explanation adequately addresses the reviewer's concerns, further illustrating the necessity and effectiveness of our chosen methodology in unveiling novel biomarkers with significant potential in the diagnosis and understanding of MASLD.

	COL1A1	COL1A2
IGFBP7	0.57	0.70
SSC5D	0.80	0.87
SEMA4D	0.58	0.72
TNC	0.21	0.26
PLAU	0.41	0.58
CXCL10	0.28	0.37
THBS1	0.49	0.61
PAM	0.22	0.40
VCAN	0.42	0.57
ADAMTS2	0.68	0.75
FBN1	0.56	0.72

Table 2. Correlation coefficients of candidate biomarker genes with COL1A1 and COL1A2 expression in human MASLD patients.

“In 1.5 and 1.6, I suggested that the final three remaining proteins (IGFBP7, SSC5D, SEMA4D) should be described in terms of their ability to diagnose fibrosis on their own (not combined), including cutoff values. However, the authors did not answer this question. There are many papers out there that use machine learning to improve the diagnostic performance of diseases using existing parameters. If the authors would like to argue that they have found a new biomarker by narrowing down the parameters using D2O labeling in this study, they must have sufficient fibrosis diagnostic power without machine learning”.

Author response: We recognize that our previous answer may have been as clear as intended. The individual values of the biomarkers (not combined) are provided in the manuscript figure 6A, B and C for the testing cohort and figure 7A, B and C for the validation cohort, showing their mean values and the statistical differences on group level.

To address the reviewer's remark (on the ability to diagnose fibrosis on their own) we assessed the predictive capability of each biomarker individually using ANOVA's F-test in the testing cohort. The F-test is a statistical method that yields the F-statistic (in this study not to be confused with the Kleiner fibrosis stage). The F-statistic is calculated as the ratio of between-group variability to within-group variability (Table 3A). Consequently, a higher F-statistic indicates a greater likelihood of the biomarker to accurately distinguish between different groups. Additionally, this test provides a p-value for the F-statistic, offering insight into the statistical significance of the biomarker's predictive power.

In addition, we determined the cut-off values per biomarker and their individual predictive power without machine learning (Table 3B). If the biomarker concentration is below cutoff value 1, it indicates that the patient is F0/F1; if the concentration of the biomarker is between cutoff value 1 and cutoff value 2, the patient is categorized as F2; if the concentration of the biomarker is above cutoff value 2, the patient is categorized as F3/F4. Based on these cutoff values we calculated the predictive power of each of the individual biomarkers.

This indicates that the biomarkers have individual predictive value (taking into account that an accuracy of 0.33 is comparable to flipping a coin in a 3-group classification). Furthermore, we wish to highlight that an improvement is achieved by combining the individual biomarkers using a ML model, which results in a much better prediction, which is described in the manuscript.

ML models have proven to be superior in diagnosing a variety of diseases more accurately than traditional statistical methods. For example, Churpek et al. (*Critical Care Medicine* 44(2): 368-374, 2016) found that ML methods, including random forests, outperformed logistic regression models in clinical disease prediction. Additionally, ML models have been effectively utilized in diagnosing and predicting chronic diseases, indicating their potential for integration into clinical practices to improve diagnostic precision and patient outcomes (Battineni et al. *J. Pers. Med.* 2020, 10(2), 21). This includes tools for predicting the development of future cardiovascular events using a multi-panel of four biomarkers and ML, as reported by Neumann (*Biomark Med.* 2020 Jun;14(9):775-784).

A)

Biomarker	F-statistics	p-val
IGFBP7 (ng/ml)	14.168	4.70E-06
SEMA4D (ug/ml)	16.497	8.42E-07
SSC5D (ng/ml)	4.622	1.23E-02

B)

Biomarker	Cutoff value 1	Cutoff value 2	Accuracy on training set (N=90)	Accuracy on test set (N=38)	Accuracy on validation cohort (N=156)
IGFBP7 (ng/ml)	204.85	265.03	0.644	0.658	0.583
SSC5D (ng/ml)	2.479	5.356	0.756	0.711	0.622
SEMA4D (ug/ml)	0.810	1.660	0.656	0.474	0.365

Table 3. A) F-test to calculate accuracy to distinguish groups based on biomarker. B) Cut-off values of the individual biomarkers and their ability to predict fibrosis based on their serum concentrations.

“In addition, the validation set is small, with only 156 cases, and particularly, there are only 30 cases of F2. While the validation results are crucial to demonstrating the usefulness of these findings, unfortunately, the number is not sufficient”.

Author response: We would like to emphasize that the primary focus of our current research was on the identification and validation of biomarkers through a translation from pre-clinical discovery to clinical applicability. This objective shaped our methodology and guided our selection of cohort sizes. Upon the reviewer’s earlier recommendation, we have incorporated an additional independent validation cohort. We have shown in this independent validation cohort that the sample size is well suited to achieve significant effects on group-level and to perform accurate patient predictions, which, as indicated in our results (Figure 7 of the manuscript), exceeds the predictive accuracy of existing NITs.

“In reply 1.7, the authors argued that fibrosis was an acceptable endpoint. Since there have been many papers claiming to have discovered novel fibrosis markers, a biomarker that predicts only fibrosis is not novel. If the authors do not know whether a novel fibrosis marker has the ability to predict liver-related events including decompensation events and cancers, then an extremely high level of fibrosis diagnostic ability should be required. At each stage, the diagnostic performance must be statistically significantly higher than that of existing fibrosis markers, such as FIB-4 index”.

Author response: To identify effective biomarkers for fibrosis, we followed an innovative identification approach, different from others, leading to the discovery of novel biomarkers not previously identified. This was achieved by using methodology in preclinical models that directly links

the process of matrix deposition to the outcome, namely the discovery of unique biomarkers overlooked by previous research.

We agree that fibrosis is a known risk factor for developing liver-related complications such as hepatocellular carcinoma (HCC) and decompensated cirrhosis, and that stages F3-F4 are precursors to these serious conditions. Our study demonstrates that our biomarkers accurately capture F3-F4 stages (AUC=0.84, figure 6 and 7 in the manuscript) across two independent cohorts. It is well-documented that a significant majority (approximately 80%) of HCC cases arise from cirrhosis (stage F4). However, our diagnostic markers are not designed to diagnose HCC or decompensated cirrhosis directly; instead, they aim to detect fibrosis at an earlier stage, enabling the application of targeted therapies. Our data indicate that our biomarkers can predict F3-F4 stages more effectively than other tests, such as FIB-4, which has been shown to be an independent factor associated with increased HCC risk among NASH cirrhosis patients (Albhaisi et al., 2023). The study by Albhaisi et al. focuses on a cohort of 157 NASH, cirrhosis, and HCC patients, with a specific emphasis on FIB-4 and the occurrence of HCC/decompensated events. Recently it was also demonstrated that FIB-4 and FibroScan have prospective value for these endpoints (published by Mozes et al (Lancet Gastroenterol Hepatol. 2023;8:704-713 and co-authored by Holleboom, Mak and van Dijk)). We would like to point out that this meta-analysis was designed for the purpose by collecting the data of 25 studies including over 2,500 patients with nearly 150 events.

Given our promising results in fibrosis prediction, it is plausible that our panel could outperform FIB-4 in predicting HCC and decompensation. However, exploring this hypothesis would require a different study design.

At the reviewer's request we performed an additional analysis to compare FIB-4 and TLM3, our novel biomarker set on the complete set of clinical cohorts in this study (training set, first validation set and second validation set). The results of this analysis are shown in Table 4 below. The left part of the table presents the confusion matrix, and the right part the corresponding performance matrices (sensitivity, specificity and precision). FIB-4 is shown in Table 4A, and TLM3 shown in Table 4B. The data show that for stages F2 and F3/F4 for all performance matrices TLM3 outperformed FIB-4. For F0/F1 the specificity and sensitivity between the two biomarkers were almost similar (0.70 vs. 0.69; 0.83 vs. 0.82; FIB-4 slightly better) whereas for precision TLM3 clearly exceeded FIB-4. These results are added to the manuscript as supplemental table 4.

A)

FIB4 confusion matrix		Predicted class			Performance metrics		
		F0/1	F2	F3/4	Sensitivity	Specificity	Precision
True class	F0/1	59	21	4	0.70	0.83	0.52
	F2	32	29	7	0.43	0.75	0.28
	F3/4	23	55	31	0.28	0.64	0.74

B)

TLM3 confusion matrix		Predicted class			Performance metrics		
		F0/1	F2	F3/4	Sensitivity	Specificity	Precision
True class	F0/1	51	16	7	0.69	0.82	0.76
	F2	8	19	11	0.50	0.87	0.36
	F3/4	8	18	56	0.68	0.78	0.76

Table 4. Confusion matrices and performance metrics of FIB4 (A) and TLM3 (B) and of all samples used in the study.

Reviewer #2 (Remarks to the Author):

“Thank you for addressing the concerns raised; this has significantly improved the overall consistency of the manuscript. However, there are still some issues that need to be addressed as follows”:

“2.10 How was the data normalized before presenting it to the LightGBM model?”

Reviewer’s own Response: While it is partially true that the model’s results on the dataset used in this study might not be significantly altered by scaling/normalizing the data, the purpose of data scaling is to ensure that the methods developed here can be extended to many other datasets. These external datasets might have different data distributions, due to even minor differences in measurement techniques. The thresholds learnt by the proposed model may not produce the reported performance in such cases. One of the objectives of developing an ML model is to show that it has generalizability and data normalization is essential for that. Moreover, the results presented here need to consider the effect of this normalization. For instance, if a feature is normalized to a standard normal distribution, the parameters for this transformation are estimated from the training set (independently per fold). The validation/test sets are transformed using the learnt parameters before inference. This provides a realistic estimate of how the model is likely to generalize to other datasets.

Author Response: *We agree that the purpose of the study is to develop a generic model that can be applied to many other datasets/studies. This indeed includes that minor differences in measuring techniques should be addressed before the data is used in the model. In the testing cohort consisting of 128 samples, we did not normalize the data before it was used in the LGBM model, as indicated in our previous response. However, in the additional validation cohort of 156 samples which was analyzed on request of the reviewers, we normalized the data by adjusting by the mean of the validation set with respect to the mean of the training set. This adjustment has been included in the Methods section of the manuscript. In future unknown datasets, the variation in data due to measuring techniques will be addressed through the inclusion of reference samples.*

“2.11 -The full list of features used to train the models is not clearly specified. If both clinical and gene expression were used as features, how were they integrated?”

Reviewer’s own Response: Thanks for the clarification. However, please address the concern raised in 2.18 regarding the LDA computations.

Author Response: *See response to 2.18*

“2.16 - It is very surprising that adding sex and BMI reduces the model's performance so significantly. Given that this is a decision tree-based ensemble model, shouldn't the model be able to pick the right features even when some additional features are provided?”

Reviewer’s own Response: This necessitates the need to show if the model is biased towards a particular gender. This can be clarified by reporting the model’s performance stratified by gender.

Author Response: *After careful analysis of stratification by gender, we observed no significant differences in the accuracy of the model in predicting fibrosis groups between male and female*

(Table 5), suggesting no bias towards a particular gender. This fits with the observation that adding gender in a forced way to the model did not improve the model's performance. (supplemental figure 2 of the manuscript).

Gender	subgroup	Total	Accuracy
female	0/1	30	0.63
female	2	15	0.53
female	3/4	29	0.72
male	0/1	36	0.75
male	2	15	0.27
male	3/4	31	0.61

Table 5. LGBM model accuracy per gender in the independent validation cohort.

“2.18 - Line 252: It is unclear how the LDA features were computed and used. Were they computed independently per fold? While the LDA features were deemed important they do not show up in Fig. 5. It is surprising the LightGBM model were unable to learn the features learnt using LDA. What was the performance without using LDA?”

Reviewer's own Response: Thanks for the clarifications. In this case, the 10-fold CV results are biased due to leaky pre-processing, as the entire training set, including the validation sets used in CV, was used to compute the LDA features. This is akin to including the validation/test set to select (learn) features and then evaluate the feature's performance on the validation/test set.

Author response: The reviewer is correct in pointing out that there could be a potential for leaky pre-processing. We trained the LDA model using the training set in the characterization of the testing cohort (referred to as the testing set in the manuscript). In addition, this concern did not apply to the additional analysis such as the prediction of fibrosis score for the independent validation using the Danish cohort. Therefore, the potential problem identified by the reviewer does not arise in this context. We now have added this information to the methods section to clarify this for the reader.

“2.20 Suggestion: Since there are multiple sources of data (murine, human) and several methods of measurement (transcriptome, liver lipid analysis, proteomics etc.) as well as several analysis (DEG analysis, correlation analysis, ML-based modelling etc.), it would be good to have a block diagram that describes the connections between these.

For clarification we generated an overview of the steps taken in the project to highlight multiple sources of data and methods of measurements. If the reviewer and editor like this graphical abstract, we can add this as a figure in the supplements. “

“Reviewer's own Response: Yes, please add the figure as it enhances clarity in the manuscript.”

Author Response: thank you, we have added the figure as supplement to the manuscript.

REVIEWERS' COMMENTS

Reviewer #1 (Remarks to the Author):

The authors have responded appropriately to the reviewer's comments, and the manuscript has been improved. There are no remaining issues of particular concern.

Reviewer #2 (Remarks to the Author):

The authors have addressed most comments adequately - just a few minor comments to revise as below

Response to 2.11:

It should be clearly described that the testing cohort is actually part of a "discovery cohort". Because the entire discovery cohort was used for feature selection (LDA) there was no need for data normalization. While the authors have mean centered the data for the validation cohort, it is unclear why variance normalization is ignored. Does the model perform poorly if variance is normalized? If so, please indicate in supplementary why that is the case.

Response to 2.16:

This is a useful result and should be included in the supplementary material (in addition to Fig. 2 of the manuscript) as this shows the variability of the model across subgroups. Specifically, the accuracy for F2 in males is poor. These should be mentioned as the model's limitations in the discussion section also.

Response to 2.18:

Thanks for the clarification. The training set and testing cohort (testing set) together then should be clearly described as the "discovery cohort/discovery set" since this entire set is used for feature selection.

Dear reviewers,

We are grateful to the reviewers for their insightful and constructive comments,. Their detailed feedback which have significantly contributed to the improvement of our manuscript. We thank them for their time and expertise.

REVIEWERS' COMMENTS

Reviewer #1 (Remarks to the Author):

The authors have responded appropriately to the reviewer's comments, and the manuscript has been improved. There are no remaining issues of particular concern.

Author response: We greatly appreciate the reviewer's insightful comments and advice aimed at improving our manuscript.

Reviewer #2 (Remarks to the Author):

The authors have addressed most comments adequately - just a few minor comments to revise as below

Author response: We thank the reviewer for his/her valuable insights and constructive feedback. The few minor comments below are taken into account in the final version of the manuscript.

Response to 2.11:

It should be clearly described that the testing cohort is actually part of a "discovery cohort". Because the entire discovery cohort was used for feature selection (LDA) there was no need for data normalization. While the authors have mean centered the data for the validation cohort, it is unclear why variance normalization is ignored. Does the model perform poorly if variance is normalized? If so, please indicate in supplementary why that is the case.

Author response: We acknowledge the concern about not performing variance normalization. However, we opted to retain the original data scale as it carries biologically significant variance, which we believe is crucial for accurate predictions and analysis. This decision was further supported by our observations in the validation cohort, where the median of all biomarkers was higher compared to the training cohort, despite having a similar composition in terms of fibrosis stage groups. We attributed this difference primarily to assay or experimental settings, leading us to median center the validation data. Nonetheless, given that the origin of variance could be both biological and experimental, we maintain that preserving the original scale is beneficial for the integrity of the analysis.

With regards to the description of the testing cohort we have added the following to the methods section: "Since this cohort is used for feature selection and the development of the machine learning model, the testing cohort should also be considered a discovery cohort."

Response to 2.16:

This is a useful result and should be included in the supplementary material (in addition to Fig. 2 of the manuscript) as this shows the variability of the model across subgroups. Specifically, the accuracy for F2 in males is poor. These should be mentioned as the model's limitations in the discussion section also.

Author response: We added the table to supplemental figure 3 as suggested by the reviewer and mentioned this in the discussion section.

Response to 2.18:

Thanks for the clarification. The training set and testing cohort (testing set) together then should be clearly described as the "discovery cohort/discovery set" since this entire set is used for feature selection.

Author response: We have added a sentence in the description of the cohort that the testing cohort is used for feature selection and the development of the machine learning model. Therefore, the testing cohort should also be considered a discovery cohort.